# RLIP: Relational Language-Image Pre-training for Human-Object Interaction Detection

**Hangjie Yuan**[1*] **Jianwen Jiang**[2*] **Samuel Albanie**[3]
**Tao Feng**[2] **Ziyuan Huang**[4] **Dong Ni**[1†] **Mingqian Tang**[2]

[1]Zhejiang University    [2]Alibaba Group    [3]University of Cambridge
[4]National University of Singapore
{hj.yuan, dni}@zju.edu.cn    sma71@cam.ac.uk    ziyuan.huang@u.nus.edu
{jianwen.jjw, shisi.ft, mingqian.tmq}@alibaba-inc.com

## Abstract

The task of Human-Object Interaction (HOI) detection targets fine-grained visual parsing of humans interacting with their environment, enabling a broad range of applications. Prior work has demonstrated the benefits of effective architecture design and integration of relevant cues for more accurate HOI detection. However, the design of an appropriate pre-training strategy for this task remains underexplored by existing approaches. To address this gap, we propose *Relational Language-Image Pre-training* (RLIP), a strategy for contrastive pre-training that leverages both entity and relation descriptions. To make effective use of such pre-training, we make three technical contributions: (1) a new **Par**allel entity detection and **Se**quential relation inference (ParSe) architecture that enables the use of both entity and relation descriptions during holistically optimized pre-training; (2) a synthetic data generation framework, Label Sequence Extension, that expands the scale of language data available within each minibatch; (3) mechanisms to account for ambiguity, Relation Quality Labels and Relation Pseudo-Labels, to mitigate the influence of ambiguous/noisy samples in the pre-training data. Through extensive experiments, we demonstrate the benefits of these contributions, collectively termed RLIP-ParSe, for improved zero-shot, few-shot and fine-tuning HOI detection performance as well as increased robustness to learning from noisy annotations. Code will be available at `https://github.com/JacobYuan7/RLIP`.

## 1 Introduction

Driven by improvements in storage, sensors and networking technology, humanity is amassing vast archives of image and video data. A significant fraction of this media is *human-centric*—it is content focused on humans and their actions. The task of *human-object interaction (HOI) detection* [4] aims to provide a step towards fine-grained parsing of such content by detecting all possible triplets of the form *<human, relation, object>* present in visual data. Robust HOI detection has myriad uses for image/video data analysis and represents essential functionality for visual and language applications such as image/video captioning [62, 48], image retrieval [27], image synthesis [26] and video action understanding [24, 65].

Given that sustained progress in object detection has yielded increasingly robust systems for detecting people and objects [53, 16, 10], a key remaining challenge for HOI detection is to develop methods capable of generalising to the many possible pairs of interactions between these entities when provided with non-exhaustive training data. To tackle this challenge, we draw inspiration from recent

---

*Equal contribution. This work was done when Hangjie Yuan was an intern at DAMO Academy, Alibaba Group, supported by Alibaba Research Intern Program.
†Corresponding author.

developments demonstrating that contrastive language-image pre-training can induce remarkable generalisation for zero-shot classification tasks [52, 25]. These methods perform classification by casting it as a *retrieval problem*, ensuring that the downstream task *aligns closely* with the pre-training objective. Recent work by Alayrac et al. [2] hypothesises that it is this close alignment between the downstream and pre-training objectives that explains why contrastive methods have proven so effective for zero-shot classification. In light of this hypothesis, in this work, we explore whether it is possible to achieve a similarly close alignment between the HOI detection task and its pre-training strategy.

While HOI detection has been widely studied [15, 49, 61, 40, 29, 31, 69, 71, 30, 57, 66, 9], the topic of designing pre-training to reflect the final task objective remains under-explored. Indeed, a widely adopted strategy [66, 68, 57, 7, 71] has been to employ object detection pre-training to initialise the parameters of the model responsible for both entity detection and relation inference. However, while suitable for entity detection, such pre-training may be suboptimal for the detection of *relations between entities* which often requires the model to take account of groups of entities with greater spatial context, rather than individual entities in isolation.

To address this shortcoming of HOI detection, we propose **Relational Language-Image Pre-training** (RLIP) which tasks the model with establishing correspondences from both entities and relations to free-form text descriptions. By doing so, RLIP endows the model with the ability to perform zero-shot HOI detection[3]. Moreover, in contrast to previous pre-training schemes that are limited to predefined finite category lists, RLIP benefits from the rich descriptive nature of natural language supervision.

We encountered three barriers to a naive implementation of RLIP for existing methods: (1) Recent end-to-end HOI detection architectures [57, 68, 71, 66] typically employ joint representations of (some subset of) *subject, object* and *relation* triplets. As a consequence, it is difficult to leverage text descriptions for separate humans, objects and relations provided by existing datasets such as VG [32]. (2) Contrastive pre-training requires negative samples to train effectively, but it is unclear *a priori* how such negatives should be constructed. (3) Free-form text descriptions exhibit label noise and semantic ambiguity (since there can be many ways to describe the same concept in the absence of a canonical list of categories), rendering optimisation challenging.

To overcome these barriers, we make several technical contributions in addition to the RLIP framework. First, to allow end-to-end contrastive pre-training with distinct descriptions of subsets, objects and relations, we propose the **Par**allel entity detection and **Se**quential relation inference (ParSe) architecture. ParSe employs a DETR [3]-like design that allocates separate learnable query groups for subject and object representations, together with an additional set of conditional queries that encode relations. While ParSe enables (and works best with) RLIP, we also find that it yields gains for traditional object detection pre-training schemes. To address the second barrier, we synthesise label sequences by extending in-batch labels with out-of-batch sampling to ensure a plentiful supply of negatives—we term this Label Sequence Extension (LSE). For the third barrier, we exploit cross-modal cues to resolve label noise and relation ambiguity. In particular, to mitigate label noise we use the quality of the visual entity detection phase [37] to assign quality scores to relation-text correspondences, an approach we term Relational Quality Labels (RQL). To mitigate relation ambiguity, we leverage similarities between labels to propagate relations via a pseudo-labeling scheme, which we term Relational Pseudo-Labels (RPL).

We demonstrate through experiments that relational pre-training outperforms traditional object detection pre-training schemes on comparable data. We further find that the a zero-shot application of our combined approach, RLIP-ParSe, surpasses several existing fine-tuned methods.

## 2 Related Work

**Human-object interaction detection.** There is a rich body of work on HOI detection. One theme has focused on the development of effective architectures for this task [40, 29, 57, 9]. A second theme has sought to leverage informative cues ranging from interaction points [40, 69], interaction boxes [29], contextualized embeddings [57, 30, 71, 30], poses [15, 39] and statistical priors [31, 66] to external knowledge in the form of language embeddings [61, 49, 66]. However, relational pre-training and open-vocabulary recognition remains underexplored. The inter-pair transformations of

---

[3]*Zero-shot* in this context refers to HOI detection without fine-tuning (following the terminology of [52]), a formulation that assesses the generalization of a pre-training model to unseen distributions. This offers a practical alternative to the scenario (unseen combinations) considered in several prior HOI detection papers [19, 21].

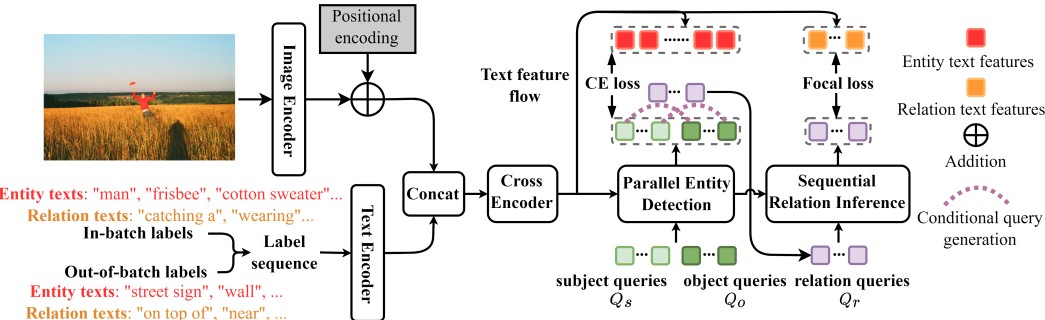

Figure 1: An overview of our pre-training framework, RLIP-ParSe. The *Parallel Entity Detection* and *Sequential Relation Inference* blocks represent independent DETR-style decoders responsible for entity and relation detection, respectively. We omit the localisation loss for clarity. See Sec. 3 for further details.

IDN [38] and affordance transfer learning of ATL [20] can be interpreted as entity augmentations to train a stronger verb classifier, but these methods do not directly optimise for all the components of HOI detection. Similar to ParSe, CDN [68] explores disentangled embeddings. However, it still couples the embedding of subjects and objects, rendering it suboptimal for RLIP. The concurrent work, GEN-VLKT [41] aims to derive knowledge from image-level language-image pre-training [52], while we aim to achieve aligned entity- and relation-level pre-training for HOI detection.

**Leveraging free-form text for visual pre-training.** A series of recent papers have illustrated the significant value of employing free-form language to provide supervision for vision systems. CLIP [52] and ALIGN [25] demonstrated striking improvements in zero-shot image classification ability through contrastive training of image-level representations. Further work has sought to additionally leverage correspondences between objects/regions and text to learn flexible grounding models [28, 36, 67]. We similarly seek to benefit from natural language supervision. However, to the best of our knowledge, we are the first to leverage correspondences between descriptions of relations and explicit pairings of *subjects and objects* (rather than descriptions of images, objects or regions) as a pre-training signal.

## 3 Methodology

In this section, we first present our triplet detection architecture, ParSe. Second, we describe how ParSe is used to perform relational language-image pre-training (RLIP). Finally, we introduce techniques to synthesise contrastive negatives and mitigate noise and ambiguities among labels. The overall RLIP-ParSe framework is illustrated in Fig. 1.

### 3.1 ParSe for Triplet Detection

**Structure overview.** The core idea underpinning the ParSe architecture is to allocate distinct representations of subjects, objects and relations in a holistically optimized model (rather than representing their combination, as commonly pursued in prior work [57]). The motivation for doing so is two-fold: (i) distinct representations enable the direct use of contrastive RLIP, since these representations can be put in correspondence with separate entity and relation annotations; (ii) the separation of responsibilities allows for a more fine-grained control over the context available for each decision (a theme that has proven important for detection tasks [56]). In particular, note that when detecting subjects and objects, local context is typically most useful. However, when it comes to relations, detection will benefit not only from informative local cues, but also neighbouring context [64] (for instance, it is useful to be aware of *water* and *hoses* when inferring the relation in the triplet *<human, wash, car>*). To instantiate this idea we follow [68] and implement triplet detection in a two-stage end-to-end manner. Our probabilistic model factorises as follows:

$$\mathbb{P}(\boldsymbol{G}|\boldsymbol{Q}_s, \boldsymbol{Q}_o, \boldsymbol{C}; \boldsymbol{\theta}_{Par}, \boldsymbol{\theta}_{Se}) = \mathbb{P}(\boldsymbol{B}_s, \boldsymbol{B}_o|\boldsymbol{Q}_s, \boldsymbol{Q}_o, \boldsymbol{C}; \boldsymbol{\theta}_{Par}) \cdot \mathbb{P}(\boldsymbol{R}|\boldsymbol{B}_s, \boldsymbol{B}_o, \boldsymbol{C}; \boldsymbol{\theta}_{Se}) \qquad (1)$$

where $\boldsymbol{Q}_s, \boldsymbol{Q}_o \in \mathbb{R}^{N_Q \times D}$ define two sets of independent queries for $N_Q$ subjects and $N_Q$ objects; $\boldsymbol{C}$ denotes features from the detection encoder; $\boldsymbol{B}_s, \boldsymbol{B}_o, \boldsymbol{R}$ denote sets of detected subject boxes, object boxes and relations, respectively (these collectively comprise the detection results $\boldsymbol{G}$); $\boldsymbol{\theta}_{Par}$ and $\boldsymbol{\theta}_{Se}$ represent learnable parameters from the entity detection decoder and the relation inferring decoder, respectively. To construct ParSe, we design two components, *Parallel Entity Detection* and *Sequential Relation Inference*, to implement the second and third terms in Eq. (1), respectively. These are described next.

**Parallel Entity Detection.** Following the DETR family of architectures [3, 70, 43], we first extract visual features using an image encoder, add positional encodings and then pass the result through a customized Transformer encoder according to the detector we adopt (we explore both DETR [3] and DDETR [70] variants) to obtain detection features $C$. Then, two sets of queries $Q_s$ and $Q_o$ are fed into the entity decoder to perform self-attention [58], cross-attention and feed-forward network (FFN) inference, obtaining $\tilde{Q}_s, \tilde{Q}_o \in \mathbb{R}^{N_Q \times D}$ which are used to predict box locations and classes.

**Sequential Relation Inference.** To encode relations, we perform *Sequential Relation Inference* as a sequential step after entity detection (similarly to [68]). In the first stage, subjects and objects are detected via *Parallel Entity Detection*. In the second stage, we adopt a simple parameter-free matching scheme between subjects and objects to generate relation queries: matching by their indices. Using this pairing scheme, we obtain relation queries via a conditional query generation function:

$$Q_r = F_{so}(\tilde{Q}_s, \tilde{Q}_o) \tag{2}$$

where for simplicity, we adopt addition as the query generation function. Since we match by indices, $Q_r \in \mathbb{R}^{N_Q \times D}$ contains $N_Q$ relation queries. $Q_r$ is then fed into the second decoder to perform *Sequential Relation Inference* via self-attention, cross-attention and FFN inference to obtain the corresponding relation features $\tilde{Q}_r \in \mathbb{R}^{N_Q \times D}$ which are then used for relation classification.

### 3.2 RLIP-ParSe for Relational Language-Image Pre-training

For each iteration of pre-training, we construct a minibatch of images and their annotated relation triplets comprising all entities' locations, $N_E$ unique entity text labels as well as $N_R$ unique relation text labels. We describe how these are used for contrastive pre-training next.

**Formation of target label sequences.** We construct targets from in-batch labels (free-form text descriptions forming subject, object, relation triplets). In more detail, we first aggregate all entity labels within the batch and append to this sequence a *no objects* label. Next, we similarly aggregate all in-batch relation labels. Then, all entity and relation labels are respectively fed into a text encoder (RoBERTa [44] in our implementation) to extract label features denoted as $L_E$ and $L_R$, respectively. Note that a free-form text label can have multiple tokens after tokenization—we use only the feature derived from the [CLS] token to represent the label. We concatenate the label feature sequence with features from the image encoder as shown in Fig. 1. To fuse the concatenated features, we adopt a simple approach: applying a Transformer encoder [34, 28, 1] to obtain fused label features $\tilde{L}_E \in \mathbb{R}^{N_E \times D}$ and $\tilde{L}_R \in \mathbb{R}^{N_R \times D}$.

**Cross-modal alignment through classification.** To implement RLIP, we task the model with establishing correspondences between entities/relations and their text descriptions using a classification objective, following [36]. In particular, we align the $i$th relation $\tilde{Q}_r(i) \in \mathbb{R}^D$ with relation its text via a Focal loss [42]:

$$P_r(i) = \tilde{Q}_r(i)\tilde{L}_R^T + \tilde{Q}_r(i)W_b^T + W_c; \quad \mathcal{L}_r(j) = \text{Focal}(\text{sigmoid}(P_r(i,j))) \tag{3}$$

where $\tilde{Q}_r(i)W_b^T + W_c$ is the learnable bias term introduced in [42]; $W_b \in \mathbb{R}^{N_R \times D}$ is a learnable linear projection and $W_c \in \mathbb{R}^{N_R}$ is a constant vector filled with $-\log((1-\pi)/\pi)$ with $\pi = .01$. The Focal loss is defined via $\text{Focal}(p) = -(1-p)^\gamma \log(p)$ where $\gamma$ is set as a hyperparameter. In the argument to this loss in Eq. (3), $j$ indexes along $P_r(i) \in \mathbb{R}^{N_R}$. To encourage matching of subjects and objects with their corresponding entity descriptions, an analogous objective to Eq. (3) is used except that a softmax and a CE loss are applied and $W_c$ is omitted (note that entities are uni-label and relations multi-label, as defined by the downstream task). The central benefit of the RLIP objectives defined above is that they bring the pre-training and downstream HOI detection losses into close alignment since the task of classifying entities and relations in the downstream task reflects the same matching task used in pre-training. As a result, RLIP produces models that can perform HOI detection under *zero-shot with no fine-tuning* (NF) evaluation protocols.

**Label Sequence Extension (LSE).** Within a given batch, the number of negative samples available for matching is limited. However, provision of plentiful negatives has been widely shown to improve contrastive learning [8, 11, 23, 59]. To this end, we propose Label Sequence Extension as a mechanism to leverage out-of-batch text descriptions. Concretely, we sample additional text descriptions with a ratio of two thirds entity labels and one third relation labels. To ensure computational tractability in the presence of the quadratic complexity of Transformer, we limit the label sequence to a predefined length $N_L$. We experiment with two sampling strategies: (i) *Uniform sampling* that draws among candidate labels with equal probability; (ii) *Frequency-based sampling* that samples according to the label frequency in the training set.

### 3.3 Addressing Relational Semantic Ambiguity

Datasets with crowd-sourced language annotations [32, 33] exhibit significant label noise and ambiguity. First, the descriptions themselves may be noisy (inaccurate), particularly when the underlying image is challenging to interpret. A second challenge for traditional training schemes is that similar relations can be described differently, thanks to synonyms. For example, the *stand near* relation may be annotated "stand near", "stand next to", "stand by", *etc.*. These forms of *semantic ambiguity* make supervised cross-modal pre-training (which relies on access to consistent labels) challenging. To mitigate this issue, we focus on two aspects of the pre-training input data: (i) the quality of the relation text labels; (ii) the presence of semantically-similar labels in sampled label sequences.

**Relational Quality Labels (RQL).** To tackle the first challenge, we propose a label smoothing [46] approach. The key idea is that we expect the difficulty of subject and object detection for a particular instance to correlate with the confidence of the annotated relation. We therefore propose to estimate annotation quality from the quality of the entity detection stage. Drawing inspiration from the generalised focal loss [37], we instantiate this idea by assessing the quality of the $i$th subject and object detection after bipartite matching [57] as

$$e(i) = \min(\text{GIoU}_{0\text{--}1}(\boldsymbol{B}_s(i), \hat{\boldsymbol{B}}_s(i)), \text{GIoU}_{0\text{--}1}(\boldsymbol{B}_o(i), \hat{\boldsymbol{B}}_o(i))) \tag{4}$$

where $\text{GIoU}_{0\text{--}1}$ denotes generalized IoU from [55] together with a linear scaling function to scale the GIoU value to the range of 0 to 1 and $\hat{}$ denotes ground-truth annotation. The resulting value $e(i)$ is then employed to calibrate the relation label confidence via multiplication: $\tilde{\boldsymbol{R}}(i) = e(i)\hat{\boldsymbol{R}}(i)$.

**Relational Pseudo-Labels (RPL).** To address the second issue, we propose a pseudo-labelling strategy [63] to account for synonyms in the extended sequence. We exploit the fact that text embeddings with high semantic similarity will lie close together, as measured by an appropriate distance function $M(\cdot, \cdot)$. We define the distance between the $i$th annotated relation label $\hat{\boldsymbol{R}}(i) = \{0, 1\}^{N_R}$ and the $j$th relation text feature $\tilde{\boldsymbol{L}}_R(j) \in \mathbb{R}^D$ from the extended sequence as

$$M(\hat{\boldsymbol{R}}(i), \tilde{\boldsymbol{L}}_R(j)) = \sum_{k=1}^{N_R} \hat{\boldsymbol{R}}(i, k) \cdot m(\tilde{\boldsymbol{L}}_R(k), \tilde{\boldsymbol{L}}_R(j)) \tag{5}$$

where $m(\cdot, \cdot)$ denotes Euclidean distance. Given the $i$th relation label, we apply a scaling function to $M(i, j)$ via $\bar{M}(i, j) = \frac{\max_k(M(i,k)) - M(i,j)}{\max_k(M(i,k))}$ where we have abbreviated $M(\hat{\boldsymbol{R}}(i), \tilde{\boldsymbol{L}}_R(j))$ as $M(i, j)$ for clarity. Next, we use a global threshold $\eta$ to select label texts with high similarities: we set the $j$th label in the $i$th relation labels as $\bar{M}(i, j)$ if $\bar{M}(i, j) > \eta$. Note that when applying either RQL or RPL, the ground truth labels are continuous (rather than discrete). We therefore employ the Quality Focal Loss [37] (rather than the standard Focal Loss [42]) as our objective function.

### 3.4 Pre-training, Fine-tuning and Inference

By design, our pre-training (RLIP) and fine-tuning phases follow a similar process. For a given batch of images with corresponding annotations, we aggregate the results from *Parallel Entity Detection* and *Sequential Relation Inference* to form $N_Q$ triplets per image. During pre-training and fine-tuning, we employ bipartite matching similarly to prior work [57, 71, 6, 68, 66], following in particular the matching cost proposed in [57]. The overall loss is then constructed as follows:

$$\mathcal{L} = \lambda_1 \mathcal{L}_{l1} + \lambda_2 \mathcal{L}_{GIoU} + \lambda_3(\mathcal{L}_s + \mathcal{L}_o) + \lambda_4 \mathcal{L}_r \tag{6}$$

where $\mathcal{L}_{l1}, \mathcal{L}_{GIoU}, \mathcal{L}_s, \mathcal{L}_o, \mathcal{L}_r$ denote the $\ell_1$ loss for box regression, GIoU loss [55], CE loss for subject and object classes, and Focal loss for relations (or Quality Focal loss [37] when applying RQL or RPL), respectively. The $\lambda$ terms are fixed weights to balance multi-task training following [57], with $\lambda_1 = 2.5, \lambda_2 = 1, \lambda_3 = 1, \lambda_4 = 1$. During pre-training, the label sequence is constructed from both in-batch and out-of-batch labels. During fine-tuning, we use all text labels contained in the dataset to form the label sequence (unlike pre-training, these labels fall within a pre-defined category list of limited size). Note that we follow [57, 68] and exclude $\mathcal{L}_s$ during fine-tuning since HOI detection detects only humans as subjects. During inference, the confidence score for an object is simply the top-1 score from the softmax distribution over objects, and the relation score is obtained by multiplying the original score from the sigmoid function and the object score. We rank relation scores and filter out the top-$K$ within those correctly localised triplets (IoU > 0.5) for evaluation. $K$ is set to 100 by default following [57, 68, 40, 66].

## 4 Experiments

**Datasets.** We use the Visual Genome (VG) [32] dataset for RLIP. This dataset contains 108,077 images annotated with free-form text for a wide array of objects and relations (100,298 object

annotations and 36,515 relation annotations). The dataset is pre-processed prior to use (see supplementary material for details). For downstream tasks, we conduct experiments on HICO-DET [5] and V-COCO [14]. HICO-DET contains 37,536 training images and 9,515 testing images, annotated with 600 HOI triplets derived from combinations of 117 verbs and 80 objects. We evaluate under the **Default** setting. V-COCO comprises 2,533 training images, 2,876 validation images and 4,946 testing images annotated with 24 interactions and 80 objects. Results are assessed under two scenarios denoted as $AP_{role}^{\#1}$ and $AP_{role}^{\#2}$ as defined by the official evaluation code [14]. Note that all object classes in HICO-DET and V-COCO are identical to COCO.

**Implementation details.** The basic architecture of the encoder and decoder are based on DETR [3] for **ParSe** and DDETR [70] for an additional variant named **ParSeD**. A detailed architecture description of RLIP-ParSeD (which uses an additional transformer for cross-modal fusion) is provided in the supplementary. For *Parallel Entity Detection* and *Sequential Relation Inference*, 3 decoding layers are used. The number of queries $N_Q$ is set to 100 during pre-training and 64 during fine-tuning (following [68]). $\gamma$ in the Focal loss is set to 2 following [57, 68]. $N_L$ in LSE is set to 500 to ensure computational tractability. $\eta$ in RPL is set to 0.3. For pre-training and fine-tuning, the initial learning rate (LR) of the image and text encoders is set to 1e-5, while all other modules are set to 1e-4. For RLIP-ParSeD and ParSeD (object detection and relation detection pre-training), we pre-train model on VG for 50 epochs and drop LR by a factor of 10 at epoch 40. For ParSeD and RLIP-ParSeD, We fine-tune for 60 epochs and drop LR at epoch 40 by a factor of 10. For ParSe and RLIP-ParSe, We fine-tune for 90 epochs and drop LR at epoch 60 by a factor of 10. The pre-training and fine-tuning strategy follow above descriptions unless stated otherwise. Experiments are conducted on 8 Tesla V100 GPU cards with a minibatch size of 32.

**Experimental protocols.** To assess performance under *fine-tuning* and *zero-shot with no fine-tuning* (NF) scenarios, we evaluate on HICO-DET across three HOI sets under the mAP metric: *Rare* (HOIs with training samples less than 10, of which there are 138), *Non-Rare* (HOIs with samples equal to or more than 10, of which there are 462) and *Full* (all HOIs, of which there are 600).

To evaluate performance under the zero-shot formulation considered by [20, 21, 19], we report results on unseen combinations (UC). In particular, we report results under two settings: *UC with rare-first* (UC-RF) selection and *UC with non-rare first* selection (UC-NF), both of which are assessed across three subsets: Unseen (120 HOIs), Seen (480 HOIs) and Full (600 HOIs).

To evaluate few-shot transfer performance, we follow [28] and sample subsets of training annotations from HICO-DET. In detail, we sample $1\%$ and $10\%$ of the total annotations available among the HICO-DET training data, ensuring that all objects and verbs (but not all combinations) exist in the selected annotations. Similarly to the fine-tuning protocol, we evaluate on the *Rare*, *Non-Rare* and *Full* sets.

To assess the robustness of RLIP and its sensitivity to noise in the relation labels, we follow [35, 60] and artificially inject noise into the relation labels by randomly flipping a fixed ratio of verbs in HOI triplets across the training set.

## 4.1 Results and Analysis

**Comparing object detection and relation detection pre-training with RLIP.** An assessment of the benefits of object detection pre-training using the COCO dataset may offer a somewhat optimistic evaluation of this approach, since COCO shares identical object classes with the downstream HOI detection evaluation datasets HICO-DET and V-COCO (and in the latter case shares training images). If we control for this effect by performing object detection pre-training on VG rather than COCO, we observe a significant drop in performance for our ParSeD baseline (from 29.12 to 23.78 across the *Full* set on HICO-DET and from 61.8 to 41.4 on V-COCO for $AP_{role}^{\#1}$, as shown in Tab. 1). However, since COCO lacks relation annotations, we investigate the benefits of RLIP on VG. We observe that RLIP outperforms vanilla object detection pre-training by a wide margin (boosting performance from 23.78 to 29.21 across the *Full* set on HICO-DET and from 41.4 to 53.1 on V-COCO for $AP_{role}^{\#1}$), demonstrating the value of incorporating relations as a pre-training cue. Another way to pre-train with relations is to perform relation detection which is still inferior to RLIP (27.45 < 29.21), demonstrating the importance of relational language-image pre-training.

**Leveraging off-the-shelf object detection data without relations.** As shown in the previous experiment, while VG provides relation annotations, it provides a much weaker basis for object

Table 1: Comparisons with previous fine-tuned methods on HICO-DET and V-COCO (column $AP^{\#1}_{role}$ and $AP^{\#2}_{role}$). PT, PTP, OD, RD and MD abbreviate Pre-Training, Pre-Training Paradigm, Object Detection, Relation Detection and Modulated Detection. FRCNN, R, HG and Swin-T denote Faster R-CNN [54], ResNet [17], Hourglass [47] and Swin-Tiny [45]. * denotes RLIP is performed on VG, initialized with parameters from COCO object detection. † denotes fine-tuning for 150 epochs following QAHOI [6].

| PT Data | PTP | Method | Detector | Backbone | Rare | Non-Rare | Full | $AP^{\#1}_{role}$ | $AP^{\#2}_{role}$ |
|---|---|---|---|---|---|---|---|---|---|
| - | - | QAHOI† [6] | DDETR | Swin-T | 22.44 | 30.27 | 28.47 | - | - |
| | - | ParSeD† | DDETR | Swin-T | **25.76** | **31.84** | **30.44** | - | - |
| COCO | OD | InteractNet [13] | FRCNN | R50-FPN | 7.16 | 10.77 | 9.94 | 40.0 | - |
| | | GPNN [51] | FRCNN | R152-DCN | 9.34 | 14.23 | 13.11 | 44.0 | - |
| | | iCAN [12] | FRCNN | ResNet-50 | 10.45 | 16.15 | 14.84 | 45.3 | 52.4 |
| | | UnionDet [29] | RetinaNet | R50-FPN | 11.72 | 19.33 | 17.58 | 47.5 | 56.2 |
| | | PPDM [40] | CenterNet | HG104 | 13.97 | 24.32 | 21.94 | - | - |
| | | HOTR [30] | DETR | ResNet-50 | 17.34 | 27.42 | 25.10 | 55.2 | 64.4 |
| | | HOITransformer [71] | DETR | ResNet-50 | 16.91 | 25.41 | 23.46 | 52.9 | - |
| | | QPIC [57] | DETR | ResNet-50 | 21.85 | 31.23 | 29.07 | 58.8 | 61.0 |
| | | OCN [66] | DETR | ResNet-50 | 25.56 | 32.51 | 30.91 | 64.2 | 66.3 |
| | | CDN [68] | DETR | ResNet-50 | 27.39 | 32.64 | 31.44 | 61.7 | 63.8 |
| | | QAHOI [6] | DDETR | ResNet-50 | 18.06 | 28.61 | 26.18 | - | - |
| | | ParSeD | DDETR | ResNet-50 | 22.23 | 31.17 | 29.12 | 61.8 | 64.0 |
| | | ParSe | DETR | ResNet-50 | 26.36 | 33.41 | 31.79 | 62.5 | 64.8 |
| | | ParSe | DETR | ResNet-101 | **28.59** | **34.01** | **32.76** | **64.4** | **66.5** |
| VG | OD | ParSeD | DDETR | ResNet-50 | 19.59 | 25.03 | 23.78 | 41.4 | 43.0 |
| | RD | ParSeD | DDETR | ResNet-50 | 21.36 | 29.27 | 27.45 | 51.5 | 53.2 |
| | RLIP | RLIP-ParSeD | DDETR | ResNet-50 | **24.45** | **30.63** | **29.21** | **53.1** | **55.0** |
| COCO+VG | RLIP* | RLIP-ParSeD | DDETR | ResNet-50 | 24.67 | 32.50 | 30.70 | 61.7 | 63.8 |
| | RLIP* | RLIP-ParSe | DETR | ResNet-50 | **26.85** | **34.63** | **32.84** | **61.9** | **64.2** |
| GoldG+ | MD | MDETR-ParSe [28] | DETR | ResNet-101 | 22.91 | 31.07 | 29.19 | 53.6 | 56.0 |

detection pre-training than COCO. More broadly, we may expect that object annotations are likely to be more readily available (and greater in scale) than relation annotations. To mitigate this, a simple solution is to simply load object detection parameters pre-trained from an object-annotated dataset (like COCO), to complement the abilities of RLIP. Tab. 1 indicates that RLIP-ParSeD indeed benefits from this approach, surpassing both object detection pre-training and RLIP ($29.12, 29.21 \rightarrow 30.70$ on the *Full* set). We pre-trained on DETR for 150 epochs, outperforming an expert object detection pre-training ($31.79 \rightarrow 32.84$). On V-COCO, there is a degradation of performance ($62.5 \rightarrow 61.9$) which we believe may be caused by the reduced domain alignment (i.e. common training images) relative to COCO object-detection pre-training [57].

**Comparing cross-modal regional alignment pre-training with RLIP.**  We next compare to the use of language-region alignment pre-training introduced by MDETR [28], which employed the GoldG+ dataset (this comprises VG, COCO and Flickr30k [50] together with the corresponding annotations for referring expressions, VG regions, Flickr entities, and GQA [22]). For comparison, we initialise RLIP-ParSe with MDETR's parameters and then fine-tune on HICO-DET. The results are reported in Tab. 1 as MDETR-ParSe. Although MDETR makes use of a heavier backbone (ResNet-101) and additional pre-training data, RLIP-ParSe nevertheless surpasses this baseline with a lighter ResNet-50 backbone, demonstrating the effectiveness of RLIP for this task.

**Zero-shot HOI detection.**  To assess performance under the zero-shot NF protocol, we compare with other methods using RLIP-ParSe (initialised with COCO parameters followed by RLIP on VG) and ParSe (initialised with COCO parameters). Note that during RLIP, we match subjects against a diverse collection of categories. However HOI detection only needs to detect *person* as subjects. Consequently, for zero-shot NF inference, we filter out subjects that are not classified as *person*. We report results in Tab. 2, where we observe that RLIP outperforms several fully fine-tuned methods. We find, as expected, that regional language-image pre-training methods like MDETR fail under an NF evaluation, since its pre-training lacks the notion of explicit relations. Under UC-RF and UC-NF protocols (fine-tuning for 40 epochs under UC-NF to avoid over-fitting), RLIP-ParSe outperforms previous methods and ParSe by performing RLIP on VG. **Few-shot transfer on HICO-DET.** To

evaluate few-show transfer, we fine-tune ParSeD for 60 epochs as above, while RLIP-ParSeD is fine-tuned for 10 epochs to avoid over-fitting. The results are shown in Tab. 3. We observe that RLIP significantly benefits few-shot fine-tuning relative to object detection pre-training and relation detection pre-training, especially when data is scarce.

Table 2: Results under zero-shot settings on HICO-DET. NR denotes Non-Rare.

| Zero-shot | Method | Rare | NR | Full |
|---|---|---|---|---|
| NF | MDETR-ParSe [28] | 0.00 | 0.00 | 0.00 |
| | RLIP-ParSe | **15.08** | **15.50** | **15.40** |

| Zero-shot | Method | Unseen | Seen | Full |
|---|---|---|---|---|
| UC-RF | VCL [19] | 10.06 | 24.28 | 21.43 |
| | ATL [20] | 9.18 | 24.67 | 21.57 |
| | FCL [21] | 13.16 | 24.23 | 22.01 |
| | ParSe | 18.53 | 32.21 | 29.06 |
| | RLIP-ParSe | **19.19** | **33.35** | **30.52** |
| UC-NF | VCL [19] | 16.22 | 18.52 | 18.06 |
| | ATL [20] | 18.25 | 18.78 | 18.67 |
| | FCL [21] | 18.66 | 19.55 | 19.37 |
| | ParSe | 19.65 | 24.50 | 23.38 |
| | RLIP-ParSe | **20.27** | **27.67** | **26.19** |

Table 3: Few-shot transfer on HICO-DET. OD, RD denote object detection pre-training and relation detection pre-training.

| Method | Data | Epochs | Rare | Non-Rare | Full |
|---|---|---|---|---|---|
| ParSeD (VG, OD) | 1% | 60 | 0.18 | 2.05 | 1.62 |
| | 10% | 60 | 6.46 | 12.19 | 10.87 |
| ParSeD (VG, RD) | 1% | 60 | 3.74 | 8.62 | 7.50 |
| | 10% | 60 | 12.29 | 17.98 | 16.67 |
| ParSeD (COCO, OD) | 1% | 60 | 5.86 | 10.16 | 9.17 |
| | 10% | 60 | 12.20 | 20.39 | 18.51 |
| RLIP-ParSeD (VG) | 0% | - | 12.30 | 12.81 | 12.69 |
| | 1% | 10 | 16.24 | 16.05 | 16.09 |
| | 10% | 10 | 15.43 | 20.34 | 19.21 |
| RLIP-ParSeD (COCO + VG) | 0% | - | 11.20 | 14.73 | 13.92 |
| | 1% | 10 | 16.22 | 18.92 | 18.30 |
| | 10% | 10 | 15.89 | 23.94 | 22.09 |

**The influence of relation label noise.** To assess sensitivity to noise, we report fine-tuning results on HICO-DET with increasing ratios of relation label noise in Fig. 2. We observe that as label noise increases, the COCO object detection pre-training adopted by prior work exhibits a greater degradation in performance (29.12→24.52, -4.60) than RLIP (29.21→25.68, -3.53) We also observe that when initialising RLIP-ParSeD with COCO pre-trained parameters, RLIP again helps to ameliorate noise, with a more limited loss of performance (30.70→26.87, -3.83) than COCO pre-training and a similar degradation to RLIP-ParSeD with random initialization. Consequently, we deduce that RLIP offers a route to mitigating label corruption and improving model robustness [18].

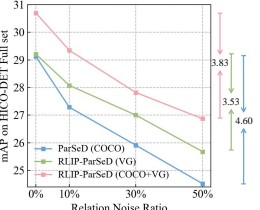

Figure 2: Relation Label Noise.

Table 4: Ablation study of different sampling strategies for label sequence extension using RLIP-ParSeD on HICO-DET.

| Sampling Type | Fine-tuning | | | Zero-shot (NF) | | |
|---|---|---|---|---|---|---|
| | Rare | Non-Rare | Full | Rare | Non-Rare | Full |
| - | 22.58 | 28.98 | 27.51 | 9.77 | 9.97 | 9.92 |
| Uniform | 23.33 | 29.55 | 28.12 | 9.46 | 9.67 | 9.63 |
| Frequency-based | 23.02 | 29.77 | 28.22 | 10.45 | 11.26 | 11.07 |

## 4.2 Ablation studies and analysis

**Ablation study of ParSe on the influence of decoupled representations.** We report an ablation study of the ParSe architecture in Tab. 5 to highlight the importance of decoupling the representation of subjects, objects and relations. The first row of Tab. 5 represents the use of coupled representations for subjects, objects and relations [57]. The second row of Tab. 5 represents the use of coupled representations for subjects and objects that are disentangled from relations [68]. The final row (ParSe) uses fully-disentangled representations. We observe a clear gain resulting from ParSe over methods using a joint representation of (some subset of) subject, object and relation triplets.

**Ablation study of sampling strategy.** In Tab. 4, we present an ablation study to assess the efficacy of out-of-batch sampling strategies. Intuitively, *uniform sampling* will up-weight descriptions from the tail of the distribution while *frequency-based sampling* will preserve the distribution. Both bring improvements to fine-tuning by providing additional negative samples. However, by over-sampling descriptions from the tail, uniform sampling performs less well with common texts in the downstream task, and thus fares less well overall.

**Visualisation of ParSe attention weights.** We visualise attention weights from ParSe for several example images in Fig. 3. We observe that ParSe attends distinct regions for subjects, objects and relations. This aligns with our motivation that entity detection is best supported by local context,

Table 5: Fine-tuning results with ParSe (COCO, OD) on HICO-DET.

| ParSe Architecture | Coupling | Rare | Non-Rare | Full |
|---|---|---|---|---|
| - | coupled subject, objects and relations | 23.18 | 31.45 | 29.55 |
| w/ Se | coupled subject and objects | 25.58 | 32.50 | 30.91 |
| w/ ParSe | fully decoupled | **26.36** | **33.41** | **31.79** |

Table 6: Ablation study of RLIP techniques using RLIP-ParSeD on HICO-DET.

| RLIP Technique | | | Fine-tuning | | | Zero-shot (NF) | | | Relation | |
|---|---|---|---|---|---|---|---|---|---|---|
| LSE | RQL | RPL | Rare | Non-Rare | Full | Rare | Non-Rare | Full | Uniformity↓ | Alignment↓ |
| | | | 22.58 | 28.98 | 27.51 | 9.77 | 9.97 | 9.92 | -0.8233 | 0.3650 |
| ✓ | | | 23.02 | 29.77 | 28.22 | 10.45 | 11.26 | 11.07 | -1.0556 | 0.4542 |
| ✓ | ✓ | | 24.32 | 30.32 | 28.94 | 11.49 | 12.60 | 12.34 | -1.3986 | 0.6072 |
| ✓ | ✓ | ✓ | 24.45 | 30.63 | 29.21 | 12.30 | 12.81 | 12.69 | -1.3265 | 0.5799 |

while relation inference draws on additional spatial context, conditioning on subjects and objects like hands and string, as well as the wet ground and sky where appropriate.

**Ablation study of RLIP techniques.** In Tab. 6, we present an ablation study of the three proposed technical contributions. We observe that each benefits both fine-tuning and zero-shot NF under all metrics. LSE attains a greater boost for the *Non-Rare* set (by sampling according to the training set distribution). On the other hand, RPL enhances results for the *Rare* set, likely due to the propensity of RPL to label rare descriptions as positive (due to the long-tailed distribution of text labels).

**Understanding LSE, RQL and RPL with Uniformity and Alignment.** To gain insight into representation quality, Wang et al. [59] proposed two metrics, *uniformity* and *alignment*, which we employ here to better understand the influence of our contributions. To this end, we perform a zero-shot (NF) evaluation on HICO-DET, using bipartite matching to assign predicted triplets to ground-truth labels. We then calculate uniformity and alignment metrics for relation features via $\mathcal{L}_u(f;t) = \log(\mathbb{E}_{(x,y)\overset{\text{i.i.d}}{\sim}\mathcal{P}_{data}}[e^{-t\|f(x)-f(y)\|_2^2}])$ and

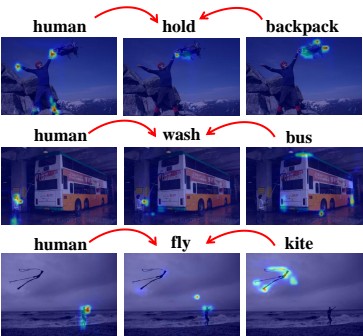

Figure 3: Attention weight analysis for the top-1 scored verb. Weights are extracted from *Parallel Entity Detection* (human and object) and *Sequential Relation Inference* (verb).

$\mathcal{L}_a(f;\alpha) = \mathbb{E}_{(x,y)\sim\mathcal{P}_{pos}}[\|f(x)-f(y)\|_2^\alpha]$, where $\alpha, t = 2$. We present the results in Tab. 6, where we observe that LSE and RQL both reduce uniformity, the former through additional negative pairs and the latter by reducing the loss assigned to over-confident text labels. We also observe that RPL yields better alignment through discovering additional positive labels. The results indicate, however, a trade-off between the metrics—a useful direction for future work would be to determine how to find an appropriate balance between them.

**Verb-wise mAP Analysis for zero-shot (NF) evaluation.** We provide analysis to give a sense of the verb overlap of HICO with VG. We use "relationship aliases" from the official VG website to obtain as many HOI verb annotations from VG as possible by string matching. The result is shown in Tab. 11 in the supplementary material. We observe that in VG there are only 2,203 HOI verb annotations even when considering relationship aliases—approximately 1.47% of the number of relationship annotations in HICO-DET. 30 HOI verbs do not have an annotation and 45 HOI verbs have five or fewer annotations. In RLIP-ParSe (COCO+VG), we observe that mAP for the 30 verbs is 5.56 while mAP for the remaining 87 verbs is 18.12. If we use uni-modal relation detection pre-training, the result for the 30 verbs degrades to zero. In light of this, we conjecture that existing relations can transfer their knowledge to the inference of non-existing relations in HOI detection. To provide a more detailed analysis, we show the verb-wise mAP on HICO verbs in VG (Fig. 3) and not in VG (Fig. 4) with zero-shot (NF) evaluation (figures are provided in the supplementary material), where we observe solid performance for some verbs.

**Probing into reasons for the verb zero-shot performance.** We aim to *qualitatively understand where the zero-shot ability stems from*. In the above analysis, *pay* has the highest performance among verbs not seen by VG (Fig. 4 in the supplementary material). In the methodology section, we

Table 7: VG verb ranking given similar subject-object triplets from HICO-DET. Verbs are in ascending order of Euclidean distance. (The Cosine distance can also output similar rankings.)

| "pay"
("parking meter") | putting money in | collecting money at | puts change into | repairing | checking | next to | ... |
|---|---|---|---|---|---|---|---|
| Count | 1 | 1 | 1 | 1 | 1 | 1 | ... |
| Euclidean | 11.56 | 11.70 | 13.34 | 14.21 | 15.16 | 16.12 | ... |
| Cosine | 0.4560 | 0.4576 | 0.3108 | 0.2554 | 0.1583 | 0.0709 | ... |

present the conditional query generation that constrains the verb inference to be related to subjects and objects, providing verb inference with a conditional context. Thus, to analyze how this ability of verb zero-shot inference emerges, we need to consider the subject and object context as they are essential to predict the verb in ParSe. For the verb *pay* in HICO-DET, there is only one possible triplet annotated, "person pay parking meter". Then, we want to answer, "**Is there any triplet annotated with similar or identical subjects and objects that transfer the inference ability to *pay*?**" To answer this question, we search for triplets annotated with similar subjects and objects to HICO-DET from VG (For details, please refer to the analysis of Tab. 12 in the supplementary material.). We report the verb distribution of the limited number of triplets that are found, ranking the verbs in ascending order of Euclidean distance to the target verb (Tab. 7). From this table, we can see that the verbs quantitatively closer (in Euclidean distance or Cosine distance) to *pay* have similar meanings to *pay*, shown by their lexical variants or grammatical variants (e.g., *putting money in* has a similar meaning to *pay*). Thus, in the VG dataset, there is *human putting money in parking meter*, which may transfer to the zero-shot recognition of *person pay parking meter* in HICO-DET. More examples can be found in Tab. 12 in the supplementary material. In short conclusion, we conjecture that the zero-shot inference ability of RLIP is not from the scale of annotations (by comparing relation detection pre-training and RLIP using VG), but the ability to transfer the verb inference knowledge from semantically similar annotations. This analysis also accords with previous works [52, 36] that semantic diversity is important as it introduces large-scale potential annotations, ensuring a model transfers well to different data distributions.

Second, we aim to ***demonstrate quantitatively how RLIP pre-trains the model to perform zero-shot detection from the perspective of representation learning***. We employ the Uniformity metric introduced in [59]. Uniformity is a metric to assess a model's generalization in contrastive learning. In this case, since label textual embeddings serve as a classifier in RLIP, we calculate

Table 8: Uniformity analysis of the seen verbs, unseen verbs and all verbs before and after RLIP. Lower uniformity value is better.

| Verb Set | Seen (87) | Unseen (30) | All (117) |
|---|---|---|---|
| Before RLIP | -0.00367 | -0.00436 | -0.00388 |
| After RLIP | -3.73780 | -3.59457 | -3.71330 |

the Uniformity of the seen verbs, unseen verbs and all verbs, aiming to observe how the generalization changes before and after RLIP, and how the generalization varies between seen verbs and unseen verbs. The results are shown in Tab. 8. As can be seen from the table, Uniformity values are high before RLIP, suggesting that the representations before RLIP are distributed compactly, leading to a poor classifier. However, after RLIP is performed, the 87 seen verbs have a substantially lower Uniformity value, corresponding with decent zero-shot performance. Similarly, the 30 unseen verbs and the combination of 117 verbs also have excellent Uniformity values, contributing to unseen zero-shot performance. Through this quantitative observation, we think that from the perspective of representations, RLIP contributes to improved zero-shot performance.

From all the above analysis, we conjecture that the zero-shot performance is not caused by the increased dataset size or annotations, but rather from the generalization in representations obtained by pre-training with language supervision.

## 5 Conclusion

In this paper, we propose RLIP as a pre-training strategy for HOI detection. We show that RLIP, together with our additional technical contributions, boosts HOI detection performance under fine-tuning, zero-shot and few-shot evaluations, and improves robustness against noisy annotations.

**Acknowledgements:** We would like to appreciate anonymous reviewers for their valuable feedback and members from Fundamental Vision Intelligence Team of Alibaba DAMO Academy for sharing computational resources. This work was supported in part by National Natural Science Foundation of China Grant No. 62173298 and by Alibaba Group through Alibaba Research Intern Program.

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
