# RLIP: Relational Language-Image Pre-training for Human-Object Interaction Detection

**Hangjie Yuan**[1]*  **Jianwen Jiang**[2]*  **Samuel Albanie**[3]
**Tao Feng**[2]  **Ziyuan Huang**[4]  **Dong Ni**[1]†  **Mingqian Tang**[2]

[1]Zhejiang University    [2]Alibaba Group    [3]University of Cambridge
[4]National University of Singapore
{hj.yuan, dni}@zju.edu.cn    sma71@cam.ac.uk    ziyuan.huang@u.nus.edu
{jianwen.jjw, shisi.ft, mingqian.tmq}@alibaba-inc.com

## A  Appendix

In this supplementary material, we first discuss the potential societal impact (Appendix A.1) and limitations (Appendix A.2) of our approach. Next, we provide further details on the architecture of RLIP-ParSeD (Appendix A.3), dataset pre-processing (Appendix A.4), phased pre-training (Appendix A.5), attention analysis (Appendix A.6) and subject-object query pairing (Appendix A.7). Finally, we provide additional experiments and analysis (Appendix A.8) and discuss our use of datasets (Appendix A.9). Codes will be publicly available upon publication.

### A.1  Potential Societal Impact

By targeting improved HOI detection, our work has the potential to bring societal and commercial benefits in medical, retail, security and sports analysis applications. However, it is inherently a dual-use technology, providing functionality with scope for abuse. For example, better HOI detection may make it easier to conduct unlawful surveillance. Moreover, due to biases present among training datasets, it is also likely that our system does not perform equally across all demographics. Therefore, we caution that our approach represents a research proof-of-concept and is not suitable for real-world usage without a rigorous evaluation of the deployment context and appropriate oversight.

### A.2  Limitations and Potential Future Works

As noted in the experiment section of the main paper, one limitation of our method is its dependence on a particular form of annotations (*i.e.* the pre-training data must be annotated with relation triplets), which are not always available, or exist at a diminished scale relative to object detection annotations. In our work, we investigated one potential solution by bootstrapping RLIP from object detection parameters to mitigate annotation scarcity. Although existing relation annotations are limited, we do not anticipate that this will remain the case. Indeed, we hope that our work will inspire future work to focus on this problem and dataset contributions will follow. Besides, we provide ways to scale up datasets as future works. For example, we could reuse a grounding dataset with entities annotated. Then, a language processing tool like spaCy [8] can be adopted as a tool to obtain their relations from captions. Even if we do not have subjects and objects but only image-caption pairs, we can combine the use of spaCy and methods like GLIP [12] to create abundant triplet annotations. Based on the analysis, we think our method is still promising and inspiring, paving a path for further research.

---

*Equal contribution. This work was done when Hangjie Yuan was an intern at DAMO Academy, Alibaba Group, supported by Alibaba Research Intern Program.

†Corresponding author.

36th Conference on Neural Information Processing Systems (NeurIPS 2022).

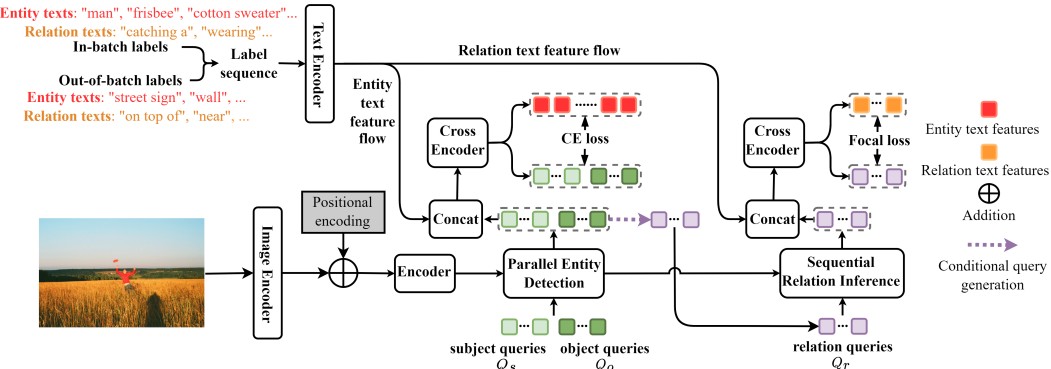

Figure 1: An overview of our pre-training framework, RLIP-ParSeD. The encoder represents DDETR-style encoders. The *Parallel Entity Detection* and *Sequential Relation Inference* blocks represent independent DDETR-style decoders responsible for entity and relation detection, respectively. The cross encoder represents DETR-style encoders. We omit the localisation loss for clarity.

A second limitation of our approach is its requirement for long optimisation schedules during pre-training—a characteristic that is inherited from DETR [2]. Although DDETR [20] ameliorates this issue to some extent, it achieves weaker performance. Thus, one potential for future work is to combine the benefits of DETR's high performance and DDETR's fast convergence speed.

## A.3 Overall Architecture of RLIP-ParSeD

We present the detailed architecture of RLIP-ParSeD as shown in Fig. 1. The major difference between RLIP-ParSe (shown in the main paper) and RLIP-ParSeD is the cross-modal fusion module. For the former, we use the detection Transformer encoder to directly fuse language-image features as previous works have done [11, 9, 1]. For the latter, we use an additional Transformer encoder to fuse the decoded queries and language features following [13] since deformable attention [20] from DDETR relies on spatial coordinates, which language features do not have. Note that the decoded subject and object queries are fused with entity text features and the decoded relation queries are fused with relation text features. The two cross encoders do not share parameters. The localisation loss is applied on decoded queries (after *Parallel Entity Detection* and before the cross encoder). Other architecture/RLIP details follow the descriptions provided in the main paper.

## A.4 Pre-Processing Steps for Visual Genome

Due to the crowd-sourcing process used to construct Visual Genome [10], there are many redundant annotations. Thus, we conduct basic cleaning steps to filter out such annotations and all of our experiments are conducted on the dataset after pre-processing. The steps are listed as follows:

- We keep the first object text description for every object because a very small proportion of objects have multiple text descriptions.
- We filter out redundant triplets by **i)** keeping only one if there are multiple identical triplets and **ii)** keeping only one if there are multiple triplets with identical subject descriptions, object descriptions, relation descriptions and similar box locations (with both the subject box's and the object box's IoU>0.5)
- We filter out redundant triplets if the number of triplets in one image is greater than the number of queries $N_Q = 100$.

## A.5 Additional Details for Phased Pre-training

In the experiments section of the main paper, we describe **i)** how object detection parameters (obtained via pre-training on COCO) can be used to initialise RLIP-ParSe and RLIP-ParSeD and **ii)** how MDETR [9] parameters (obtained via pre-training on GoldG+) can be used to initialise MDETR-ParSe. Here, we provide further details on how these are implemented.

For the main blocks of RLIP-ParSe pre-trained on COCO, we initialise the image encoder, cross encoder, *Parallel Entity Detection* block and *Sequential Relation Inference* block parameters with

parameters from the image encoder, detection Transformer encoder, detection Transformer decoder (first 3 layers) and detection Transformer decoder (first 3 layers) of a COCO pre-trained DETR model following [19]. We initialise the FFN layers of RLIP-ParSe pre-trained on COCO using the localization FFN layer parameters and entity classification FFN layer parameters (since object categories in HICO-DET and V-COCO are identical to COCO). For RLIP-ParSeD pre-trained on COCO, parameter initialisation follows RLIP-ParSe. Other parameters are randomly initialised.

For the main blocks of MDETR-ParSe [9], we initialise the parameters of the text encoder, image encoder, cross encoder, *Parallel Entity Detection* block and *Sequential Relation Inference* block with parameters from the text encoder, image encoder, cross encoder, detection Transformer decoder (first 3 layers) and detection Transformer decoder (first 3 layers) of a GoldG+ pre-trained MDETR model. For the FFN layers of MDETR-ParSe, we initialise from localization FFN layers. Other parameters are randomly initialised.

### A.6 Details of Attention Weight Analysis

In Figure 3 of the main paper, we provide an analysis of the attention weights produced by ParSe (extracted from ParSe in RLIP-ParSe). Here, we provide additional details of how this analysis is conducted. During image inference, we employ a Transformer [17] architecture to decode queries. In this decoding process, a $\mathrm{softmax}$ function is used to normalize attention weights calculated by a scaled dot-product attention. The logits after the $\mathrm{softmax}$ function indicate the importance of regions (since queries aggregate values according to the logits). Thus, we extract the logits after the $\mathrm{softmax}$ function in the last Transformer layer of both the *Parallel Entity Detection* and the *Sequential Relation Inference* block for the top-1 scored verb. To visualize the attention weights, we linearly scale the range of logits to 0–255 (and cast to integers to produce an image).

### A.7 Computational Overhead of the Subject and Object Query Pairing

The pairing of humans and objects are performed by index-matching as is stated in the main paper. Thus, we pair humans and objects with identical indices (e.g., the first decoded feature from the subject queries and the first decoded feature from the object queries are paired.). Due to the simplicity of this matching strategy, the cost is trivial ($\mathcal{O}(1)$ cost) compared to the overall overhead during model inference.

### A.8 Additional Experiments and Analysis

**Ablation study of ParSe on the influence of decoupled representations.** We report a further ablation study of the ParSe architecture in Tab. 1 to highlight the importance of decoupling the representation of subjects, objects and relations. The first row of Tab. 1 represents the use of coupled representations for subjects, objects and relations [16]. The second row of Tab. 1 represents the use of coupled representations for subjects and objects that are disentangled from relations [19]. The final row (ParSe) uses fully-disentangled representations. We observe a clear gain resulting from ParSe over methods using a joint representation of (some subset of) subject, object and relation triplets.

Table 1: Fine-tuning results with ParSe (COCO) on HICO-DET.

| ParSe Architecture | Coupling | Rare | Non-Rare | Full |
| --- | --- | --- | --- | --- |
| - | coupled subject, objects and relations | 23.18 | 31.45 | 29.55 |
| w/ Se | coupled subject and objects | 25.58 | 32.50 | 30.91 |
| w/ ParSe | fully decoupled | **26.36** | **33.41** | **31.79** |

**Robustness towards different backbones.** Compare more thoroughly with CDN [19] and QA-HOI [5], we perform extensive experiments to demonstrate the effectiveness of the uni-modal detection pipeline ParSe as shwon in Tab. 2. As the table indicates, ParSe outperforms CDN-L with half the number of decoding layers and a single-stage fine-tuning with a clear gain (+0.69 mAP on Full set). When compared to QAHOI, ParSe improves by 1.18 mAP on Full set with only two fifths number of fine-tuning epochs. If using the same number of epochs, ParSe can surpass it by 1.97 mAP on Full set and more improvement on the Rare set (+3.32mAP).

Table 2: Fully-finetuned results on HICO-DET with different backbones. PTP, DL and PT denote Pre-training paradigm, decoding layers and pre-training.

| Method | Backbone | DL | PTP | PT data | #Tuning Epochs | Rare | Non-Rare | Full |
|---|---|---|---|---|---|---|---|---|
| CDN-L [19] | ResNet-101 | 12 | OD | COCO | 90+10 | 27.19 | 33.53 | 32.07 |
| ParSe | ResNet-101 | 6 | OD | COCO | 90 | 28.59 | 34.01 | 32.76 |
| QAHOI [5] | Swin-T | 6 | - | - | 150 | 22.44 | 30.27 | 28.47 |
| ParSe | Swin-T | 6 | - | - | 60 | 23.77 | 31.40 | 29.65 |
| ParSe | Swin-T | 6 | - | - | 150 | 25.76 | 31.84 | 30.44 |

**Superiority over other models with VG and COCO data**  To have a fairer comparison with previous methods using VG and COCO data, we adopt CDN [19] as a base method and then add the VG dataset to its pre-training stage. Since RLIP also adopts the relation annotations in VG, we also try to include these annotations in the uni-modal pre-training. Thus, we resort to relation detection on VG. To be more specific, we perform uni-modal relation detection pre-training by using linear classifiers for verbs and entities rather than matching with texts. The results are shown in Tab. 3. We can see from the table that by using uni-modal relation detection pre-training, CDN still trails RLIP-ParSe with the same number of epochs of pre-training and fine-tuning, which shows the effectiveness of RLIP. Even if comparing it with ParSe using relation detection pre-training, we can still observe an improvement of ParSe over CDN, demonstrating the usefulness of decoupling triplet representations.

Table 3: Fully-finetuned results on HICO-DET with VG and COCO dataset. RD and PT denote relation detection and pre-training.

| Method | Detector | Data | PT Paradigm | PT #Epochs | Rare | Non-Rare | Full |
|---|---|---|---|---|---|---|---|
| CDN | DETR | COCO+VG | Relation Detction | 150 | 25.65 | 32.75 | 31.12 |
| ParSe | DETR | COCO+VG | Relation Detction | 150 | 26.00 | 33.40 | 31.70 |
| RLIP-ParSe | DETR | COCO+VG | RLIP | 150 | 26.85 | 34.63 | 32.84 |

**Few-shot transfer with RLIP-ParSe.**  To evaluate few-show transfer with ParSe and RLIP-ParSe, we fine-tune ParSe for 90 epochs as above, while RLIP-ParSe is fine-tuned for 10 epochs to avoid over-fitting. The detailed results are shown in Tab. 4 and evaluation curves are shown in Fig. 2. We observe that RLIP with object detection initialisation significantly benefits few-shot fine-tuning relative to object detection pre-training in terms of final results and convergence speed, especially when data is scarce. RLIP-ParSe with 1% data achieves similar performance with ParSe with 10% data.

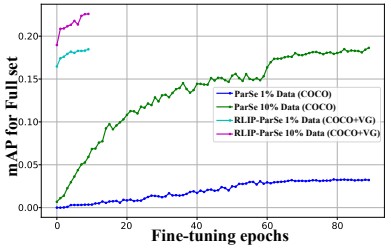

Figure 2: Evaluation curves for few-shot transfer on HICO-DET with ParSe and RLIP-ParSe.

Table 4: Few-shot transfer on HICO-DET with ParSe and RLIP-ParSe.

| Method | Data | Epochs | Rare | Non-Rare | Full |
|---|---|---|---|---|---|
| ParSe (COCO) | 1% | 90 | 1.69 | 3.67 | 3.21 |
| | 10% | 90 | 14.61 | 19.56 | 18.42 |
| RLIP-ParSe (COCO+VG) | 0% | - | 15.08 | 15.50 | 15.40 |
| | 1% | 10 | 17.47 | 18.76 | 18.46 |
| | 10% | 10 | 20.16 | 23.32 | 22.59 |

**Detailed results of relation label noise with RLIP-ParSe and RLIP-ParSeD**  We present detailed results concerning the influence of relation label noise on RLIP-ParSeD and ParSeD in Tab. 5 (which corresponds to Figure 2 in the main paper) and the influence of relation label noise on RLIP-ParSe and ParSe in Tab. 6. The RLIP-ParSe and ParSe results support our claim that RLIP helps to ameliorate noise since ParSe suffers a greater degradation of performance (31.79→25.19, -6.6) than RLIP-ParSe (32.84→27.75, -5.09).

Table 5: Relation label noise on HICO-DET with ParSeD and RLIP-ParSeD.

| Method | Noise | Rare | Non-Rare | Full |
|---|---|---|---|---|
| ParSeD (COCO) | 0% | 22.23 | 31.17 | 29.12 |
|  | 10% | 19.63 | 29.58 | 27.29 |
|  | 30% | 17.14 | 28.52 | 25.91 |
|  | 50% | 15.82 | 27.12 | 24.52 |
| RLIP-ParSeD (VG) | 0% | 24.45 | 30.63 | 29.21 |
|  | 10% | 21.59 | 30.02 | 28.08 |
|  | 30% | 19.60 | 29.21 | 27.00 |
|  | 50% | 17.11 | 28.24 | 25.68 |
| RLIP-ParSeD (COCO+VG) | 0% | 24.67 | 32.50 | 30.70 |
|  | 10% | 19.86 | 32.20 | 29.35 |
|  | 30% | 18.45 | 30.62 | 27.82 |
|  | 50% | 17.81 | 29.58 | 26.87 |

Table 6: Relation label noise on HICO-DET with ParSe and RLIP-ParSe.

| Method | Noise | Rare | Non-Rare | Full |
|---|---|---|---|---|
| ParSe (COCO) | 0% | 26.36 | 33.41 | 31.79 |
|  | 10% | 21.59 | 30.80 | 28.68 |
|  | 30% | 20.52 | 29.99 | 27.81 |
|  | 50% | 15.01 | 28.23 | 25.19 |
| RLIP-ParSe (COCO+VG) | 0% | 26.85 | 34.63 | 32.84 |
|  | 10% | 24.62 | 33.54 | 31.49 |
|  | 30% | 23.12 | 31.75 | 29.77 |
|  | 50% | 20.09 | 30.04 | 27.75 |

**Sensitivity analysis of hyper-parameter $\eta$ in RPL.** In Tab. 7, we present a sensitivity analysis for the $\eta$ hyperparamter used in RPL. As $\eta$ decreases, RPL selects more descriptions with high similarities as positive, boosting performance. However, if $\eta$ is too low, there is an increased risk of false positives arising in the pseudolabeling process. We choose $\eta = 0.3$ in the main paper according to this experiment.

Table 7: Results with varying $\eta$ in RPL with RLIP-ParSeD on HICO-DET. LSE and RQL are used by default.

| $\eta$ | Fine-tuning | | | Zero-shot (NF) | | |
|---|---|---|---|---|---|---|
|  | Rare | Non-Rare | Full | Rare | Non-Rare | Full |
| 0.2 | 24.05 | **30.73** | 29.19 | 10.95 | 12.71 | 12.30 |
| 0.3 | **24.45** | 30.63 | **29.21** | **12.30** | **12.81** | **12.69** |
| 0.4 | 23.67 | 29.90 | 28.47 | 11.97 | 12.80 | 12.61 |
| 0.5 | 22.63 | 29.79 | 28.14 | 11.70 | 12.09 | 12.00 |

**Design choice of distance function $m(\cdot, \cdot)$ in RPL.** For the design choice of $m(\cdot, \cdot)$, we also experiment on another widely-adopted distance function Cosine distance. We conduct a sensitivity analysis of the hyper-parameter $\eta$ to compare with the one chosen in the main paper (last row of results). The zero-shot (NF) results of Cosine distance using RLIP-ParSeD is shown in Tab. 8. We observe that Euclidean distance is slightly better (the last row of results is selected in the paper). Since both methods have similar computational overhead, in the paper, we choose the Euclidean distance.

Table 8: Zero-shot (NF) results with varying $\eta$ in RPL with RLIP-ParSeD on HICO-DET. LSE and RQL are used by default.

| Distance Function | $\eta$ | Rare | Non-Rare | Full |
|---|---|---|---|---|
| Cosine | 0.3 | 11.21 | 12.53 | 12.23 |
| Cosine | 0.4 | 11.92 | 12.82 | 12.61 |
| Cosine | 0.5 | 11.76 | 12.71 | 12.49 |
| Cosine | 0.6 | 11.30 | 12.22 | 12.01 |
| **Euclidean** | **0.3** | **12.30** | **12.81** | **12.69** |

**Comparing RQL and RPL with Label Smoothing Regularization.** Since we employ RPL and RQL to smooth the target distributions used during training to account for ambiguity, it is useful to compare their effectiveness to a manually-designed Label Smoothing Regularisation (LSR) method, such as the one introduced in [15]. We experiment with using LSR for relation labels (following RPL and RQL). However, we find that LSR tends to degrade performance, while the proposed RPL and RQL approaches boost all metrics.

Table 9: Results comparing Label Smoothing Regularization (LSR) [15] with RPL+RQL on HICO-DET. We use RLIP-ParSeD with LSE as a base model.

| Ambiguity suppression | Fine-tuning | | | Zero-shot (NF) | | |
|---|---|---|---|---|---|---|
| | Rare | Non-Rare | Full | Rare | Non-Rare | Full |
| - | 23.02 | 29.77 | 28.22 | 10.45 | 11.26 | 11.07 |
| LSR | 23.51 | 29.38 | 28.03 | 10.03 | 10.84 | 10.65 |
| RPL+RQL | **24.45** | **30.63** | **29.21** | **12.30** | **12.81** | **12.69** |

**Similarity analysis between in-batch labels and out-of-batch labels**   The in-batch labels are aggregated from images' annotations, and the out-of-batch labels are sampled from the whole dataset, which does not overlap with in-batch labels. Since the contrastive loss optimizes to push away the negative textual labels, we can observe the change of the similarities of the negative and positive labels. To quantitatively analyze it, we simulate the training process by out-of-batch sampling, and observe the change of similarities by calculating the average pairwise distance of the positive labels to the negative labels. We mainly compare the object and relation similarity based on the RoBERTa model before and after RLIP pre-training. The results are shown in Tab. 10. From this table, we can see that the Cosine similarity decreases, and Euclidean distance increases. Note that before RLIP, the discrimination ability of text embeddings are poor, which corresponds with previous work [6]. The results indicate whichever distance function we adopt (Cosine or Euclidean distance) and whichever kind of feature we observe (object or relation), the similarity between in-batch labels and out-of-batch labels decreases after performing RLIP. This enables the language model to adapt well to the visual representations and serve as a good classifier.

Table 10: Similarity analysis between in-batch labels and out-of-batch labels before and after RLIP. Cos and Euc abbreviate Cosine distance and Euclidean distance.

| Model | Object (Cos) | Relation (Cos) | Object (Euc) | Relation (Euc) |
|---|---|---|---|---|
| **RoBERTa (Before RLIP)** | 0.9991 | 0.9986 | 0.2502 | 0.3156 |
| **RoBERTa (After RLIP)** | 0.0084 | 0.0208 | 18.1943 | 16.9177 |

**Verb-wise mAP analysis for zero-shot (NF) evaluation**   We provide analysis to give a sense of the verb overlap of HICO with VG. We use "relationship aliases" from the official VG website to obtain as many HOI verb annotations from VG as possible. The result is shown in Tab. 11.

Table 11: Verb overlap of HICO with VG.

| Dataset | #images | HOI verb annos | HOI verb annos' ratio | Imbalance ratio |
|---|---|---|---|---|
| VG | 108K | 2,203 | 1.47% | 304 |

We observe that in VG, there are 2,203 HOI verb annotations even when considering relationship aliases—approximately 1.47% of the number of relationship annotations in HICO-DET. 30 HOI verbs do not have an annotation and 45 HOI verbs have five or fewer annotations. In RLIP-ParSe (COCO+VG), we observe that mAP for the 30 verbs is 5.56 while mAP for the remaining 87 verbs is 18.12. If we use uni-modal relation detection pre-training, the result for the 30 verbs degrades to zero.

To provide a more detailed analysis, we show the verb-wise mAP on HICO verbs in VG (Fig. 3) and not in VG (Fig. 4) with zero-shot (NF) evaluation. We can observe solid performance for some verbs.

**Probing into reasons for the verb zero-shot performance**   We aim to *qualitatively understand where the zero-shot ability stems from*. In the above analysis, *pay* has the highest performance among verbs not seen by VG. In the main paper, we present the conditional query generation that constrains the verb inference to be related to subjects and objects, providing verb inference with a conditional context. Thus, to analyze how this ability of verb zero-shot inference emerges, we need to consider the subject and object context as they are essential to predict the verb in ParSe. For the verb *pay* in HICO-DET, there is only one possible triplet annotated, "person pay parking meter". Then, we want to answer, "**is there any triplet annotated with similar or identical subjects and**

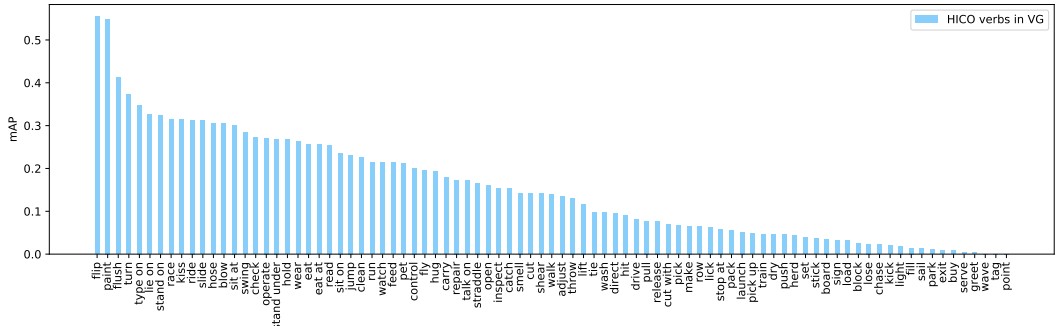

Figure 3: Verb-wise mAP Analysis for Zero-Shot (NF) Evaluation. Presented verbs exist in VG.

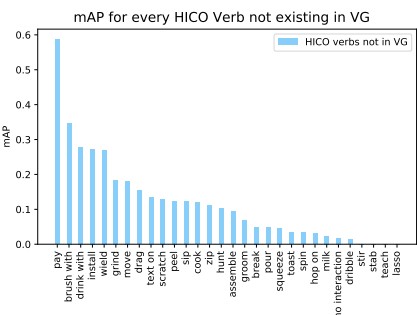

Figure 4: Verb-wise mAP Analysis for Zero-Shot (NF) Evaluation. Presented verbs do not exist in VG.

**objects that transfer the inference ability to** *pay***?**" To answer this question, we search for triplets annotated with similar subjects and objects to HICO-DET from VG. For the subjects, we heuristically select ones whose textual descriptions have any one of the following strings: *man, woman, person, friend, guy, dude, human, people, driver, passenger, hand, limb*. For the objects, we heuristically select ones whose textual descriptions have the string of the target object. By this processing, only a limited number of triplets are found. Building on these, we report the verb distribution of the limited number of triplets that are found, ranking the verbs in ascending order of Euclidean distance to the target verb, results of which are shown in Tab. 12. From this table, we can see that the verbs quantitatively closer (in Euclidean distance or Cosine distance) to *pay* have similar meanings to *pay*, shown by their lexical variants or grammatical variants (e.g., *putting money in* has a similar meaning to *pay*). Thus, in the VG dataset, there is *human putting money in parking meter*, which may transfer to the zero-shot recognition of *person pay parking meter* in HICO-DET. Similarly, we could see that in the VG datatset, there is *human putting condiments on hot dog*, which may transfer to the zero-shot recognition of *person cook hot dog* in HICO-DET. Note that there are some grammatical variations that we can not exhaustively set rules to avoid, thus creating a zero-shot setting which is not theoretically strict enough. But we want the model to benefit from this property of natural language in the context of language-image pre-training.

In short conclusion, with the assistance of the sequential inference structure of verbs, we think that the zero-shot inference ability in RLIP is not from the scale of annotations (by comparing relation detection pre-training and RLIP using VG), but the ability to transfer the verb inference knowledge from semantically similar annotations. This analysis also accords with previous papers [14, 12] that semantic diversity is important as it introduces large-scale potential annotations, ensuring a model transfers well to different data distributions.

Secondly, we aim to **demonstrate quantitatively how RLIP pre-trains the model to perform zero-shot detection from the perspective of representation learning**. We resort to the Uniformity metric introduced in [18]. Uniformity is a metric to assess a model's generalization in contrastive learning. We detail the calculation of this metric in the analysis of the main paper. In this case, since label textual embeddings serve as a classifier in RLIP, we calculate the Uniformity of the seen verbs, unseen verbs and all verbs, aiming to observe how the generalization changes before and after RLIP, and how the generalization varies between seen verbs and unseen verbs. The results are shown in Tab. 13. As can be seen from the table, Uniformity values are all high before RLIP. It means that the

Table 12: VG verb ranking given similar subject-object triplets from HICO-DET. Verbs are in ascending order of Euclidean distance. (The Cosine distance can also output similar rankings.)

**"pay"** ("parking meter")

| | putting money in | collecting money at | puts change into | repairing | checking | next to | ... |
|---|---|---|---|---|---|---|---|
| Count | 1 | 1 | 1 | 1 | 1 | 1 | ... |
| Euclidean | 11.56 | 11.70 | 13.34 | 14.21 | 15.16 | 16.12 | ... |
| Cosine | 0.4560 | 0.4576 | 0.3108 | 0.2554 | 0.1583 | 0.0709 | ... |

**"cook"** ("hot dog")

| | putting condiments on | prepping | displeased with | roasts | blowing on | about to eat | ... |
|---|---|---|---|---|---|---|---|
| Count | 1 | 1 | 1 | 1 | 1 | 1 | ... |
| Euclidean | 13.38 | 14.14 | 15.48 | 15.63 | 16.04 | 16.27 | ... |
| Cosine | 0.3467 | 0.2565 | 0.0656 | 0.1787 | 0.0471 | 0.0680 | ... |

**"grind"** ("skateboard")

| | race downhill | flying off ramp on | skating | midair on | for balancing on | competes on | ... |
|---|---|---|---|---|---|---|---|
| Count | 1 | 1 | 2 | | 1 | 1 | ... |
| Euclidean | 13.26 | 13.27 | 13.44 | 13.59 | 14.26 | 14.36 | ... |
| Cosine | 0.3670 | 0.3401 | 0.3553 | 0.3288 | 0.2510 | 0.2037 | ... |

**"assemble"** ("kite")

| | are preparing their | launched | managing | launch | carry | directing | ... |
|---|---|---|---|---|---|---|---|
| Count | 1 | 1 | 2 | 1 | 1 | 2 | ... |
| Euclidean | 12.48 | 12.72 | 13.13 | 13.20 | 13.31 | 13.71 | ... |
| Cosine | 0.3350 | 0.3684 | 0.2974 | 0.2554 | 0.2027 | 0.2565 | ... |

**"text on"** ("cell phone")

| | viewing messages on | typing on | texting | speaking on | listening on | speaks on | ... |
|---|---|---|---|---|---|---|---|
| Count | 1 | 1 | 2 | 1 | 1 | 2 | ... |
| Euclidean | 10.24 | 11.24 | 11.42 | 11.85 | 11.85 | 11.98 | ... |
| Cosine | 0.5876 | 0.4437 | 0.3983 | 0.4082 | 0.4281 | 0.3557 | ... |

**"scratch"** ("dog")

| | touching | bent over touching | touches | interacting with | holding a hot | reaching for | ... |
|---|---|---|---|---|---|---|---|
| Count | 8 | 1 | 2 | 1 | 1 | 2 | ... |
| Euclidean | 14.02 | 14.47 | 14.54 | 14.60 | 14.80 | 14.81 | ... |
| Cosine | 0.2157 | 0.1566 | 0.2260 | 0.0969 | 0.0512 | 0.2039 | ... |

representations before RLIP are compactly distributed, serving as an awful classifier. However, after RLIP is performed, the seen 87 verbs have a distinctively lower Uniformity value, corresponding with the decent zero-shot performance. Similarly, the 30 unseen verbs and the combination of 117 verbs also have excellent Uniformity values, contributing to the unseen zero-shot performance. Through this quantitative observation, we think that from the perspective of representations, RLIP contributes to the real zero-shotness.

From all the above analysis, we think that the zero-shotness may not be caused by the mounting dataset size or annotations, but stem from the generalization in representations obtained by pre-training with language supervision.

Table 13: Uniformity analysis of the seen verbs, unseen verbs and all verbs before and after RLIP. Lower uniformity value is better.

| Verb Set | Seen (87) | Unseen (30) | All (117) |
|---|---|---|---|
| Before RLIP | -0.00367 | -0.00436 | -0.00388 |
| After RLIP | -3.73780 | -3.59457 | -3.71330 |

**The influence of semantic-diverse data (upstream data distributions)** We ablate semantic diversity by significantly altering the distribution of VG annotations and assessing the influence on RLIP's performance. To this end, first note that since VG is human-annotated with free-form text, it is extremely long-tailed. We alter its distribution by dropping tail object classes and verb classes to create a dataset with limited semantic diversity. Concretely, we drop object classes whose instance counts are fewer than 1,000 and relation classes whose instance counts are fewer than 500. We

pre-train RLIP on the resulting dataset and then perform zero-shot (NF) evaluation on HICO-DET. The results are shown in Tab. 14. We observe from this table that despite a very significant change to the training distribution, performance on the Full set drops only moderately. We do, however, witness a relatively larger decline on the Rare set due to the lack of semantic diversity in the modified data. This finding accords with the observations of GLIP [12]. To make full use of language-image pre-training, semantic diversity is important which can ensure a good domain transfer as is indicated by CLIP [14] and GLIP [12].

Table 14: Semantic diversity analysis with zero-shot (NF) evaluation on HICO-DET. Obj, rel and annos denote object, relation and annotations respectively.

| Method | Data | Obj classes | Obj annos | Rel classes | Rel annos | Rare | Non-Rare | Full |
|--------|------|-------------|-----------|-------------|-----------|------|----------|------|
| ParSeD | VG | 100,298 | 3.80m | 36,515 | 1.99m | 12.30 | 12.81 | 12.69 |
| ParSeD | VG- | 497 | 1.73m | 151 | 1.27m | 9.45 | 12.13 | 11.51 |

**More successful and failure cases analysis and corresponding potential future work** In this work, we present ParSe as an effective HOI detection structure. In Fig. 5 (a-d), we present several cases where the model successfully predicts by using RLIP-ParSeD (VG), with verb scores greater than 0.3. From Fig. 5 (a) and (b), we can see that although the scene is complex with many possible triplets with multiple labels, the model can detect the right subjects and objects, linking them as triplets. From Fig. 5 (c) and (d), we can observe that although the person is only partially visible, the model still detects him/her and then predicts the right triplets. Both cases show that RLIP-ParSeD can overcome difficult cases during application. However, there are also failure cases, where further works can improve upon. We show the failure cases from top-3 predictions produced by RLIP-ParSeD (VG) in Fig. 5 (e-h). **i)** First of all, the verb inference in ParSe conditions on the detection results since Sequential Relation Inference is fed with queries generated by detection features. Also, VG does not provide a good object detection foundation. Thus, an inferior detection result can lead to false positives. As shown in Fig. 5(f) and (g), the model detects the objects to be *wine glass* and *cell phone* (which should be *fire hydrant* and *handbag*), thus producing wrong predictions. Building on this observation, we think excellent object detectors can be designed and incorporated into HOI detection. **ii)** Secondly, DETR-based models may find it hard to detect fine-grained poses in people, which degrades the performance of inferring some verbs like *wave*. As shown in Fig. 5 (e) and (g), the model fails to predict *wave* although right localizing the subject person. Building on this observation, we think it to be promising to efficiently incorporate pose cues in end-to-end HOI detection models. **iii)** Thirdly, some contextual cues may be hard to be detected. As shown in Fig. 5 (h), the model predicts *ride* rather than *lasso*. If the model captures the rope as a global context, then it can perform well. Building on this observation, we think it to be promising to efficiently incorporate fine-grained contextual cues or introduce external object knowledge to detect the triplet.

### A.9 Datasets used in this work

**Licenses.** The V-COCO [7] dataset is used under an MIT license. The HICO-DET [4, 3] dataset is used under a CC0: Public Domain license. The Visual Genome [10] dataset is used under a Creative Commons Attribution 4.0 International License.

**Release of personally identifiable information/offensive content/consent.** We do not release data as part of this research. We work with standard public domain benchmarks for computer vision: Visual Genome [10], HICO-DET [4, 3], and V-COCO [7]. We therefore assess that the risk of releasing personally identifiable information or offensive content is relatively low. With regards to consent, we did not pursue an independent investigation of consent that goes beyond the considerations of the original dataset releases.

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

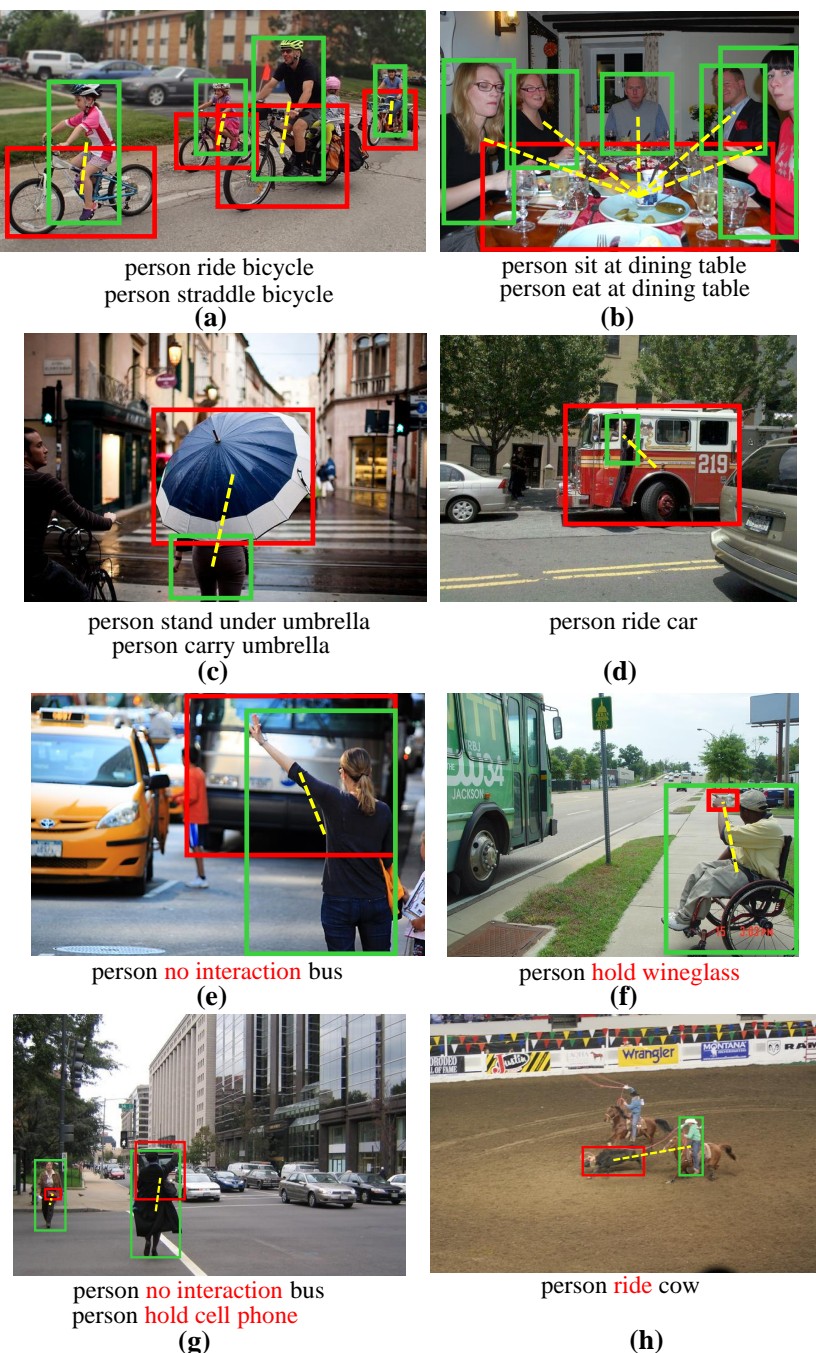

person ride bicycle
person straddle bicycle
**(a)**

person sit at dining table
person eat at dining table
**(b)**

person stand under umbrella
person carry umbrella
**(c)**

person ride car
**(d)**

person no interaction bus
**(e)**

person hold wineglass
**(f)**

person no interaction bus
person hold cell phone
**(g)**

person ride cow
**(h)**

Figure 5: Successful and failure case analysis. We visualize the true positives (a-d) with verb scores higher than 0.3, and the false positives (e-h) from top-3 predictions produced by the RLIP-ParSeD (VG) model. Words in red indicate wrong predictions.

[3] Yu-Wei Chao, Yunfan Liu, Xieyang Liu, Huayi Zeng, and Jia Deng. Learning to detect human-object interactions. In *2018 IEEE Winter Conference on Applications of Computer Vision (WACV)*, pages 381–389. IEEE, 2018.

[4] Yu-Wei Chao, Zhan Wang, Yugeng He, Jiaxuan Wang, and Jia Deng. Hico: A benchmark for recognizing human-object interactions. In *Proceedings of the IEEE International Conference on Computer Vision*, pages 1017–1025, 2015.

[5] Junwen Chen and Keiji Yanai. Qahoi: Query-based anchors for human-object interaction detection. *arXiv preprint arXiv:2112.08647*, 2021.

[6] Jun Gao, Di He, Xu Tan, Tao Qin, Liwei Wang, and Tieyan Liu. Representation degeneration problem in training natural language generation models. In *International Conference on Learning Representations*, 2019.

[7] Saurabh Gupta and Jitendra Malik. Visual semantic role labeling. *arXiv preprint arXiv:1505.04474*, 2015.

[8] Matthew Honnibal and Ines Montani. spacy 2: Natural language understanding with bloom embeddings, convolutional neural networks and incremental parsing. *To appear*, 7(1):411–420, 2017.

[9] Aishwarya Kamath, Mannat Singh, Yann LeCun, Gabriel Synnaeve, Ishan Misra, and Nicolas Carion. Mdetr-modulated detection for end-to-end multi-modal understanding. In *Proceedings of the IEEE/CVF International Conference on Computer Vision*, pages 1780–1790, 2021.

[10] Ranjay Krishna, Yuke Zhu, Oliver Groth, Justin Johnson, Kenji Hata, Joshua Kravitz, Stephanie Chen, Yannis Kalantidis, Li-Jia Li, David A Shamma, et al. Visual genome: Connecting language and vision using crowdsourced dense image annotations. *International Journal of Computer Vision*, 123(1):32–73, 2017.

[11] Junnan Li, Ramprasaath Selvaraju, Akhilesh Gotmare, Shafiq Joty, Caiming Xiong, and Steven Chu Hong Hoi. Align before fuse: Vision and language representation learning with momentum distillation. *Advances in Neural Information Processing Systems*, 34, 2021.

[12] Liunian Harold Li, Pengchuan Zhang, Haotian Zhang, Jianwei Yang, Chunyuan Li, Yiwu Zhong, Lijuan Wang, Lu Yuan, Lei Zhang, Jenq-Neng Hwang, et al. Grounded language-image pre-training. *arXiv preprint arXiv:2112.03857*, 2021.

[13] Muhammad Maaz, Hanoona Rasheed, Salman Hameed Khan, Fahad Shahbaz Khan, Rao Muhammad Anwer, and Ming-Hsuan Yang. Multi-modal transformers excel at class-agnostic object detection. *ArXiv*, abs/2111.11430, 2021.

[14] Alec Radford, Jong Wook Kim, Chris Hallacy, Aditya Ramesh, Gabriel Goh, Sandhini Agarwal, Girish Sastry, Amanda Askell, Pamela Mishkin, Jack Clark, et al. Learning transferable visual models from natural language supervision. *arXiv preprint arXiv:2103.00020*, 2021.

[15] Christian Szegedy, Vincent Vanhoucke, Sergey Ioffe, Jon Shlens, and Zbigniew Wojna. Rethinking the inception architecture for computer vision. In *Proceedings of the IEEE Conference on Computer Vision and Pattern Recognition*, pages 2818–2826, 2016.

[16] Masato Tamura, Hiroki Ohashi, and Tomoaki Yoshinaga. Qpic: Query-based pairwise human-object interaction detection with image-wide contextual information. In *Proceedings of the IEEE/CVF Conference on Computer Vision and Pattern Recognition*, pages 10410–10419, 2021.

[17] Ashish Vaswani, Noam Shazeer, Niki Parmar, Jakob Uszkoreit, Llion Jones, Aidan N Gomez, Łukasz Kaiser, and Illia Polosukhin. Attention is all you need. In *Advances in Neural Information Processing Systems*, pages 5998–6008, 2017.

[18] Tongzhou Wang and Phillip Isola. Understanding contrastive representation learning through alignment and uniformity on the hypersphere. In *International Conference on Machine Learning*, pages 9929–9939. PMLR, 2020.

[19] Aixi Zhang, Yue Liao, Si Liu, Miao Lu, Yongliang Wang, Chen Gao, and Xiaobo Li. Mining the benefits of two-stage and one-stage hoi detection. *Advances in Neural Information Processing Systems*, 34, 2021.

[20] Xizhou Zhu, Weijie Su, Lewei Lu, Bin Li, Xiaogang Wang, and Jifeng Dai. Deformable detr: Deformable transformers for end-to-end object detection. *arXiv preprint arXiv:2010.04159*, 2020.