# OpenReview forum: "RLIP: Relational Language-Image Pre-training for Human-Object Interaction Detection"
_NeurIPS.cc/2022/Conference — NeurIPS 2022 Accept_

### Official Review · Reviewer_MVep · 2022-07-04

**Rating:** 5
**Confidence:** 4
**Soundness:** 2 fair
**Presentation:** 2 fair
**Contribution:** 2 fair

**Summary:**

This work proposes a visual-language pre-training method for HOI detection. Following MDETR and CDN, a pipeline named ParSe is proposed to implement contrastive learning upon humans, objects, and relations. Correspondingly, a pos-neg language labels scheme and a language ambiguity suppression method are proposed for the pre-training. On commonly-used benchmarks, the proposed method is compared with SOTAs on both supervised and zero-shot settings.

**Questions:**

1. Efficiency of the sequential h-o pairing?

2. How similar of the pos and neg prompts? Need an analysis.

3. As the pre-training using VG, many zero-shot relations of HICO-DET maybe not zero-shot no more, as there may be many similar and even same relations in VG for training. So a detailed analysis should be taken to probe the **real "zero-shotness"**.

4. In Eq. 5, is the Euclidean distance the best choice? How about the others like cosine distance?

**Limitations:**

N/A.

**Strengths And Weaknesses:**

Pros:
+ The HOI contrastive learning is an interesting topic and would advance the HOI detection, considering the possibility of open-vocabulary visual-language learning.

+ Some reasonable model designs are proposed to handle the pos-neg label scheme, language label processing, etc.

+ Extensive ablation studies are conducted and presented. Some analysis is useful for follow-ups.

+ Borrowing the idea from MDETR into the v-l HOI contrastive learning is non-trivial.

Cons:
- First, is the novelty of the first contribution. As mentioned by the authors too, they follow CLIP, CDN, and MDETR to build the ParSe model. Though under the v-l setting, few novel designs are proposed to bring new insights.

- Second, the experiment. In Tab 1, the comparison between methods is mainly within the setting using Res-50 which is reasonable, but for CDN, some results are missing: CDN-B with Res-50: 27.55 33.05 31.78 (which is comparable with the proposed method 26.36 33.41 31.79 in COCO as PT data); and many previous works are also not compared here. If compared with the above result of CDN, it seems marginal improvement is achieved. Though performance does not mean all, considering the similar structure of the proposed method with CDN, concern raises. Meanwhile, if using Res-101 and Swin-tiny, how would ParSe perform, and does it have an advantage compared with CDN and QAHOI as the same series methods. Moreover, ParSe can use extra data in the pre-training, but few improvements are achieved.

- In the discussion of the first contribution, in fact, many previous methods (iCAN, TIN, etc) adopted the separated representations for a human, object, and relations. Even using a transformer, the differences and similarities should also be discussed. And CDN is another case, which is followed by this work. Thus, I suggest revising the part about the first contribution.
As for the relation contrastive learning, there are some recent works like Contrastive Visual and Language Translational Embeddings for Visual Relationship Detection, Unsupervised Vision-Language Parsing: Seamlessly Bridging Visual Scene Graphs with Language Structures via Dependency Relationships (just for your information, does need to discuss as they are too new).
L84: open-vocabulary recognition remains underexplored. This point is also open to discussion, as VCL, FCL,  Detecting Unseen Visual Relations Using Analogies, etc. also have considered open h-v-o scenarios.
L88: rendering it suboptimal for RLIP. Why, please give a more detailed discussion.

- Method part is somewhat hard to follow as the dense information presented is a short section. And many adopted methods are just given a citation without a brief introduction, which makes the reading interrupted.

- Prompt diversity needs a detailed analysis.

- Is the zero-shot setting fair (tab 2) as the proposed method can use extra data and the other method do not? Please clarify this.

Typo: L63 Parse --> ParSe

---

> ### Author Response · Authors · 2022-08-02
> **Response to Reviewer MVep (Part 3/3)**
>
> >**Q7**: L88: Rendering it sub-optimal for RLIP. Why, please give a more detailed discussion.
>
> **A7**: Here, what we mean by ``sub-optimal'' is that during contrastive learning, we hope the alignment of visual representations with the textual descriptions can be fine-grained and one-to-one matching as it could benefit the downstream detection task [8,9,10]. To be more specific, we design the model to align at the entity and relation levels. If we adopt previous models like QPIC [11] and CDN [12], we will fail to achieve this alignment. Besides, this alignment can benefit the detection task as it also prevents multi-task learning for a given decoded feature in a position-sensitive task like detection [13]. We present the ablation of this decoupled design to observe its contribution (also in the Supplementary Material):
>
> | ParSe Architecture | Coupling | Rare | Non-Rare | Full |
> | ----- | :----- | :-----: | :-----: | :-----: |
> | - | coupled subject, objects and relations [10] | 23.18 | 31.45 | 29.55 |
> | w/ Se | coupled subject and objects [11] | 25.58 | 32.50 | 30.91 |
> | w/ ParSe | fully decoupled | 26.36 | 33.41 | 31.79 |
>
> We can see that by decoupling the representations, we could boost performance gradually. Thus, we think that previous methods are sub-optimal for RLIP.
>
>
> >**Q8**: How similar of the pos and neg prompts? Need an analysis.
>
> **A8**: First of all, since "prompt" does not appear in our paper, we interpret the positive and negative prompts as in-batch labels and out-of-batch labels (we invite the reviewer to respond in case we have misunderstood their question.)
>
> The in-batch labels are aggregated from images' annotations, and the out-of-batch labels are sampled from the whole dataset, which does not overlap with in-batch labels. Since the contrastive loss optimizes to push away the negative textual labels, we can observe the change of the similarities of the negative and positive labels. To be more specific, we simulate the training process by out-of-batch sampling, and observe the change of similarities by calculating the average pairwise distance of the positive labels to the negative labels. We mainly compare the object and relation similarity based on the RoBERTa model before and after RLIP pre-training. The results are shown in the table below. (Cos and Euc denote using Cosine distance and Euclidean distance as a distance metric.)
>
> | Model | Object (Cos) | Relation (Cos) | Object (Euc) | Relation (Euc) |
> | ----- | :----- | :-----: | :-----: | :-----: |
> | Before RLIP | 0.9991 | 0.9986 | 0.2502 | 0.3156 |
> | After RLIP  | 0.0084 | 0.0208 | 18.1943 | 16.9177 |
>
> From this table, we can see that the cosine similarity decreases, and Euclidean distance increases. Note that before RLIP, the discrimination ability of text embeddings are poor, which corresponds with previous work [14].
> This indicates whichever distance function we adopt (Cosine or Euclidean distance) and whichever kind of feature we observe (object or relation), the similarity between in-batch labels and out-of-batch labels decreases after performing RLIP. This enables the language model to adapt well to the visual representations and serve as a good classifier.
>
> **Reference**:
> [1] Grounded language-image pre-training, CVPR 2022.
> [2] Zero-shot recognition via semantic embeddings and knowledge graphs, CVPR 2018.
> [3] Semantic relation reasoning for shot-stable few-shot object detection, CVPR 2021.
> [4] ican: Instance-centric attention network for human-object interaction detection, arXiv 2018.
> [5] Transferable interactiveness knowledge for human-object interaction detection, CVPR 2019.
> [6] Deep contextual attention for human-object interaction detection, ICCV 2019.
> [7] Learning to Detect Human-Object Interactions, WACV 2018.
> [8] MDETR-modulated detection for end-to-end multi-modal understanding, ICCV 2021.
> [9] RegionCLIP: Region-based Language-Image Pretraining, CVPR 2022.
> [10] How Much Can CLIP Benefit Vision-and-Language Tasks? ICLR 2022.
> [11] Qpic: Query-based pairwise human-object interaction detection with image-wide contextual information, CVPR 2021.
> [12] Mining the benefits of two-stage and one-stage HOI detection, NeurIPS 2021.
> [13] Revisiting the sibling head in object detector, CVPR 2020.
> [14] Representation degeneration problem in training natural language generation models, ICLR 2019.

---

> > ### Comment · Reviewer_MVep · 2022-08-03
> > **Post-rebuttal-3**
> >
> > Q7: addressed.
> >
> > Q8: addressed. Thanks for the detailed responses, please add them to the paper if possible.

---

> > ### Comment · Reviewer_MVep · 2022-08-03
> > **Overall post rebuttal**
> >
> > Overall, my main concerns are basically addressed. I appreciate the responses and efforts of the authors. Though I still have some concerns about the novelty and claim discussions, I think the revised paper would give the community some good results and inspiration. I raise my rating to 5.

---

> > > ### Author Response · Authors · 2022-08-07
> > > **Response to the Remaining Concerns (Part 3/3)**
> > >
> > > **[Question]** **With regard to the potential bias in CLIP-style models**, we think this can be caused by the collected data. Usually, to pre-train a large-scale language-image pre-training model, data is from diverse sources, and the quantity of them can also be varied. This bias will be obvious when we adapt this pre-trained model to downstream tasks especially when the task is specialized [2,3], because under this circumstance, the data distributions of upstream and downstream tasks are misaligned (also termed natural distribution shifts in CLIP [2]).
> > >
> > > One reason that text suffers more bias might be caused by the redundancy in free-form texts [4]. Thus, a reasonable way to tackle this **during upstream pre-training** is to manually filter the texts and change textual distribution manually. Due to the limited usage of language syntax in image-level language-image pre-training, it's intuitive to use filtered bag-of-words to replace the original texts [4]. (RLIP might be a good way to incorporate syntax information as it has the subject-predicate-object structure.) Also, we can perform semi-aligned learning [4,5] by adding external images without texts, which can serve as a way to align with the downstream datasets. However, we tend to think a pre-training model should be as comprehensive as possible. Thus, the misalignment can also be tackled in the downstream transfer.
> > >
> > > When **adapting the pre-trained models to downstream datasets**, there are several possible ways to overcome the potential bias from the pre-trained models. The most naive one is to perform fully fine-tuning, which is also the one adopted in this work. This paradigm adapts all the parameters to downstream tasks, aiming to transfer to their distributions, while it could be over-parameterized when the downstream dataset is in a small scale. The second way is using prompt tuning [2,3,6,7], which adds a language context to the textual embeddings to align with the downstream datasets. This language context can be fine-grained and detailed natural language [2,3], continuous learnable vectors [3,6] and image-conditioned continuous learnable vectors [7]. The third way is to use adapters [8,9] to adapt features to downstream distributions. The last two methods will be two potential future works for RLIP to efficiently adapt to downstream tasks.
> > >
> > > We hope the above answer could address all your concerns. We thank you for your timely follow-ups and look forward to your reply.
> > >
> > > **Reference**:
> > > [1] https://visualgenome.org/api/v0/api_home.html
> > > [2] Learning transferable visual models from natural language supervision, ICML 2021.
> > > [3] Grounded language-image pre-training, CVPR 2022.
> > > [4] A fistful of words: Learning transferable visual models from bag-of-words supervision, arXiv 2021.
> > > [5] Self-training with Noisy Student improves ImageNet classification, CVPR 2020.
> > > [6] Learning to prompt for vision-language models, IJCV 2022.
> > > [7] Conditional Prompt Learning for Vision-Language Models, CVPR 2022.
> > > [8] Parameter-efficient transfer learning for NLP, ICML 2022.
> > > [9] Clip-adapter: Better vision-language models with feature adapters, arXiv 2021.
> > > [10] Understanding contrastive representation learning through alignment and uniformity on the hypersphere, ICML 2020.

---

> > > > ### Comment · Reviewer_MVep · 2022-08-07
> > > > **Response**
> > > >
> > > > Thanks for the reply, I think this discussion is helpful and inspiring for readers.

---

> > > ### Author Response · Authors · 2022-08-07
> > > **Response to the Remaining Concerns (Part 2/3)**
> > >
> > > ***Secondly***, we want to explore where the boost stems from by incorporating language-image pre-training. Back to the zero-shot analysis provided above, among the 30 verbs, they have diverse performances.
> > > As shown in Figure 4 in the Supplementary Material, we can observe that some verbs have decent results.
> > >
> > > In the following part, we would exemplify why zero-shot verbs can have decent performance and where the ability of zero-shot inference stems from.
> > >
> > > For example, "pay" has the highest performance among verbs not seen by VG.
> > > In the main paper, we present the conditional query generation that constrains the verb inference to be related to subjects and objects, providing verb inference with a conditional context.
> > > Thus, to analyze how this ability of verb zero-shot inference emerges, we need to consider the subject and object context as they are essential to predict the verb in ParSe.
> > > For the verb "pay" in HICO-DET, there is only one possible triplet annotated, "person pay parking meter".
> > > Then, we want to answer, "is there any triplet annotated with similar or identical subjects and objects that transfer the inference ability to 'pay'?"
> > > Aiming to answer this, we firstly find triplets annotated with similar subjects and objects in VG. For the subjects, we heuristically select ones whose textual descriptions have any one of the following strings: *man, woman, person, friend, guy, dude, human, people, driver, passenger, hand, limb*. For the objects, we heuristically select ones whose textual descriptions have the string: *parking meter*.
> > > By this processing, only 13 triplets are selected. Building on these, we report the verb distribution of the selected triplets. We rank the verbs in ascending order of Euclidean distance of this verb to "pay". (The Cosine distance can also output similar rankings.)
> > >
> > > | Verb | putting money in | collecting money at | puts change into | repairing | checking | next to | leaning | ... |
> > > | ----- | :-----: | :-----: | :-----: | :-----: | :-----: | :-----: | :-----: | :-----: |
> > > | Count | 1 | 1 | 1 | 1 | 1 | 1 | 1 | ... |
> > > | Euclidean | 11.56 | 11.70 | 13.34 | 14.21 | 15.16 | 16.12 | 16.13 |... |
> > > | Cosine | 0.4560 | 0.4576 | 0.3108 | 0.2554 | 0.1583 | 0.0709 | 0.0165 | ... |
> > >
> > > We append more examples in Table 12 of the Supplementary Material to demonstrate this phenomenon is prevailing. From this table, we can see that the verbs quantitatively closer (in Euclidean distance or in Cosine distance) to "pay" have similar semantic meanings to "pay", shown by their lexical variants or grammatical variants (e.g., "putting money in" has similar meanings to "pay"). In short conclusion, with the assistance of the sequential inference structure of verbs, we think that the zero-shot inference ability in RLIP is not from the scale of annotations, but the ability to transfer the verb inference knowledge from semantically similar annotations. This analysis also accords with previous papers [2,3] that semantic diversity is important as it introduces large-scale potential annotations, ensuring a model transfers well to different data distributions.
> > >
> > > ***Thirdly***, to demonstrate quantitatively how RLIP pre-trains the model to perform zero-shot, we resort to the Uniformity metric introduced in [10]. Uniformity is a metric to assess a model's generalization in contrastive learning. We detail the calculation of this metric in Line 349 of the main paper. In this case, since label textual embeddings serve as a classifier in RLIP, we calculate the Uniformity of the seen verbs, unseen verbs and all verbs, aiming to observe how the generalization changes before and after RLIP, and how the generalization varies between seen verbs and unseen verbs. The results are shown in the table below (Lower is better):
> > >
> > > | Verb Set | Seen (87) | Unseen (30) | All (117) |
> > > | ----- | :-----: | :-----: | :-----: |
> > > | Before RLIP | -0.00367 | -0.00436 | -0.00388 |
> > > | After RLIP | -3.73780 | -3.59457 | -3.71330 |
> > >
> > > As can be seen from the table, Uniformity values are all high before RLIP. It means that the representations before RLIP are compactly distributed, serving as an awful classifier. However, after RLIP is performed, the seen 87 verbs have a distinctively lower Uniformity value, corresponding with the decent zero-shot performance. Similarly, the 30 unseen verbs and the combination of 117 verbs also have excellent Uniformity values, contributing to the unseen zero-shot performance. Through this quantitative observation, we think that from the perspective of representations, RLIP contributes to the real zero-shotness.
> > >
> > > From all the above analysis, we think that the zero-shotness may not be caused by the mounting dataset size or annotations, but stem from the generalization in representations obtained by pre-training with language supervision.

---

> > > > ### Comment · Reviewer_MVep · 2022-08-07
> > > > **Response**
> > > >
> > > > Thanks, this analysis helps a lot to probe the performance improvement.

---

> > > > > ### Author Response · Authors · 2022-08-08
> > > > > **New Paper Revision**
> > > > >
> > > > > Dear Reviewer MVep,
> > > > >
> > > > > Thanks for your appreciation for this detailed analysis. We have revised our Supplementary Material to include a new part **Probing into reasons for the verb zero-shot performance**, starting from Line 176. Hopefully, we will include part of this analysis into the additional page of the main paper if it is accepted. We hope the current revision has addressed all your concerns, and we'd appreciate it if you also think it further improves upon the last version.
> > > > >
> > > > > Yours sincerely,
> > > > > Authors of Paper226

---

> > > ### Author Response · Authors · 2022-08-07
> > > **Response to the Remaining Concerns (Part 1/3)**
> > >
> > > We are encouraged that you raised the score, and we thank you for your appreciation for RLIP.
> > > We think RLIP can contribute to the interaction/relation detection task as CLIP does to the retrieval/classification task and GLIP does to the object detection task.
> > > With respect to your suggestions in the last response, we do the following editing (We put most of them temporarily in the Supplementary Material, but we will squeeze part of them to the main paper if it is accepted because of the additional one page.)
> > > 1. we add the Computational Overhead of the Subject and Object Query Pairing in the Supplementary Material.
> > > 2. We add the Design Choice of distance function in the Supplementary Material.
> > > 3. We add the Robustness towards different backbones in the Supplementary Material.
> > > 4. We add fairer comparisons of Zero-shot detection results in the Table 2 of the main paper.
> > > 5. We add Similarity Analysis Between In-batch Labels and Out-of-batch Labels in the Supplementary Material.
> > >
> > > **[Concern1]** **With respect to the remaining concern about the contribution claim**, to present it more precisely and distinguish it from previous two-stage methods, we make some revisions in Line 11, Line 54, Line 62 and Line 107, aiming to emphasize the contribution in the family of holistically optimized models.
> > >
> > > **[Concern2]** **With respect to the discussion about real zero-shotness**, we appreciate your insights. Indeed, analyzing the data leakage may pave the path for potential research. Thus, we provide further analysis:
> > >
> > > ***First of all***, we want to explore whether the boost stems from the mounting dataset size or the quantity of dataset annotations. Since it is a bit intractable for uni-modal pre-training models to perform zero-shot (NF) evaluation, we mainly present fully fine-tuning results. To answer this question, we need to conduct experiments to control the usage of annotations as well. Thus, we conduct uni-modal relation detection pre-training that also adopts relation annotations and then compare the fully fine-tuned results:
> > >
> > > | Method | Detector | Data | PT Paradigm | PT \#Epochs | Rare | Non-Rare | Full |
> > > | ----- | :-----: | :-----: | :-----: | :-----: | :-----: | :-----: | :-----: |
> > > | ParSeD | DDETR | VG | OD | 50 | 19.59 | 25.03 | 23.78 |
> > > | ParSeD | DDETR | VG | Relation Detction | 50 | 21.36 | 29.27 | 27.45 |
> > > | RLIP-ParSeD | DDETR | VG | RLIP | 50 | 24.45 | 30.63 | 29.21 ||
> > >
> > > Comparing OD and relation detection pre-training, we the observe relatively better performance of relation detection pre-training (23.78->27.45) because of the usage of relation annotations. It can also be interpreted as a more aligned pre-training with the downstream task. However, it's still inferior to RLIP (27.45 < 29.21 on Full and a wider gap on Rare 21.36 < 24.45).
> > > RLIP and relation detection pre-training both incorporate relations as a pre-training signal, while the latter is sub-optimal due to the semantic similarity of the free-form text labels, which can be solved by pre-training with language supervision [2,3].
> > >
> > > Recall the fact we provided in the last response: among the 2,203 HOI verb annotations contained in VG, 30 HOI verbs do not have an annotation. Note that we use strict string matching to find annotations that may omit some grammatical variations of words, but this is a common phenomenon in natural language, and it is a bit intractable to avoid all variations. While the zero-shot results on HICO-DET indicate that mAP for the 30 verbs is 5.56, and mAP for the remaining 87 verbs is 18.12. If we are using a uni-modal pre-training (object detection or relation detection), we will fail to predict the verbs (all verbs or unseen verbs) without external information introduced.
> > >
> > > | Dataset | \#images | HOI verb annos | HOI verb annos' ratio | Imbalance ratio |
> > > | ----- | :-----: | :-----: | :-----: | :-----: |
> > > | VG | 108K | 2,203 | 1.47\% | 304 |
> > >
> > > The above facts indicate that language-image pre-training still possesses its superiority over uni-modal pre-training even if controlling the variable of dataset size and annotations.

---

> ### Author Response · Authors · 2022-08-02
> **Response to Reviewer MVep (Part 2/3)**
>
> >**Q4**: Performance comparisons with CDN. Results with ResNet101 (CDN) and Swin-Tiny (QAHOI).
>
> **A4**: With respect to CDN, the performance of CDN-B (27.55 33.05 31.78) using 12 decoding layers in total, which is twice as large as our method. Thus, in the paper, we compare ParSe with CDN-S that has 6 decoding layers.
> Also, to compare more thoroughly with CDN and QAHOI, we perform more experiments to demonstrate the effectiveness of the uni-modal detection pipeline ParSe (in the below, PT denotes pre-training).
>
> | Method | Backbone |\#Decoding layers | PT paradigm | PT data | \#Tuning epochs | Rare/Non-Rare/Full |
> | ----- | :-----: | :-----: | :-----: | :-----: | :-----: | :-----: |
> | CDN-L | ResNet101 | 12 | OD | COCO | 90+10 | 27.19 / 33.53 / 32.07 |
> | ParSe | ResNet101 | 6 | OD | COCO | 90 | 28.59 / 34.01 / 32.76 |
> | QAHOI | Swin-T | 6 | - | - | 150 | 22.44 / 30.27 / 28.47 |
> | ParSe | Swin-T | 6 | - | - | 60 | 23.77 / 31.40 / 29.65 |
> | ParSe | Swin-T | 6 | - | - | 150 | 25.76 / 31.84 / 30.44 |
>
>
> As the table indicates, ParSe outperforms CDN-L with half the number of decoding layers and a single-stage fine-tuning with a clear gain (+0.69 mAP on Full set). When compared to QAHOI, ParSe improves by 1.18 mAP on Full set with only two fifths number of fine-tuning epochs. If using the same number of epochs, ParSe can surpass it by 1.97 mAP on Full set and more improvement on the Rare set (+3.32mAP).
>
> >**Q5**: Is the zero-shot setting fair (tab 2) as the proposed method can use extra data and the other methods do not?
>
> **A5**: To clarify the improvements yielded by our model, we conduct more experiments with ParSe pre-trained on COCO for a fair comparison. We present the results below and also update them into the paper.
>
> | Zero-shot | Method | Data | Unseen | Seen | Full |
> | ----- | :-----: | :-----: | :-----: | :-----: | :-----: |
> | UC-RF | VCL | COCO | 10.06 | 24.28 | 21.43 |
> | UC-RF | ATL | COCO | 9.18 | 24.67 | 21.57 |
> | UC-RF | FCL | COCO | 13.16 | 24.23 | 22.01 |
> | UC-RF | ParSe | COCO | 18.53 | 32.21 | 29.06 |
> | UC-RF | RLIP-ParSe | COCO+VG | 19.19 | 33.35 | 30.52 |
> | UC-NF | VCL | COCO | 16.22 | 18.52 | 18.06 |
> | UC-NF | ATL | COCO | 18.25 | 18.78 | 18.67 |
> | UC-NF | FCL | COCO | 16.22 | 18.52 | 18.06 |
> | UC-NF | ParSe | COCO | 19.65 | 24.50 | 23.38 |
> | UC-NF | RLIP-ParSe | COCO+VG | 20.27 | 27.67 | 26.19 |
>
> From this table, we observe that ParSe still outperforms previous methods by a significant margin. By performing RLIP building on ParSe, the relationship knowledge benefits seen combinations more. We conjecture that after fine-tuning on the downstream datasets, the model gradually loses the compositionality of language, leaving more relation inference ability instead to boost the seen performance.
>
> >**Q6**: Many previous methods (iCAN [4], TIN [5], etc) adopted the separated representations for a human, object, and relations, which needs analysis.
>
> **A6**: The motivation of our paper is to align the pre-training stage with the downstream fine-tuning with the assistance of language-image pre-training using a novel pre-training signal. Thus, to make the contrastive learning fine-grained, we propose to decouple the representations of the triplets. Although previous methods [4,5,6,7] can also have decoupled representations, they usually i) adopt off-the-shelf object detectors to extract visual features for other post-processing steps, and ii) are equipped with multiple branches. Especially the first characteristic makes it underperform, as in a complex reasoning problem, a holistic end-to-end optimized model can better adapt its features to the task itself [8]. With the help of DETR and Deformable DETR, we can leverage this insight to design ParSe, a much neater pipeline compared to previous methods [4,5,6,7]. From the perspective of performance, ParSe surpasses them by a significant margin. From the above analysis, we think that ParSe contributes a better baseline model to the research community.

---

> > ### Comment · Reviewer_MVep · 2022-08-03
> > **Post-rebuttal-2**
> >
> > Q4: Thanks for the additional results and the efforts. My concern is addressed, please add these to the paper.
> >
> > Q5: This new table depicts the comparison very clearly. Concern addressed.
> >
> > Q6: Thanks. Indeed, old detector-based two-stage methods have a different context from recent Transformer-based e2e methods. However, they are different at the implementation level instead of the theoretical level. I still suggest the authors revise the contribution claim about the disentangled representation which may cause misleading.

---

> ### Author Response · Authors · 2022-08-02
> **Response to Reviewer MVep (Part 1/3)**
>
> We thank Reviewer MVep for their valued feedback, and are encouraged that they find the relational contrastive learning valuable and non-trivial, the model design reasonable, and our ablation studies and analysis extensive. To address their concerns, we present more explanations concerning the h-o pairing (Q1), zero-shot analysis and fairer zero-shot comparisons (Q2, Q5), more experiments about the choice of distance function (Q3), more experiments to compare performance with CDN and QAHOI (Q4), contribution analysis with previous methods (Q6), the necessity of ParSe (Q7) and similarity analysis of positive and negative prompts (Q8).
>
>
> >**Q1**: Efficiency of the sequential h-o pairing.
>
> **A1**: The pairing of humans and objects are performed by index-matching as is stated in Line 132. Thus, we pair humans and objects with identical indices (e.g., the first decoded feature from the subject queries and the first decoded feature from the object queries are paired.). Due to the simplicity of this matching strategy, the cost is trivial ($\mathcal{O}(1)$ cost) compared to the overall overhead during model inference.
>
> >**Q2**: As the pre-training using VG, many zero-shot relations of HICO-DET maybe not zero-shot no more, as there may be many similar and even same relations in VG for training. So a detailed analysis should be taken to probe the real "zero-shotness".
>
> **A2**: First of all, we wish to re-clarify the meaning of zero-shot in our context. As is stated in Line 52, what we mean by zero-shot is that we assess the model without fine-tuning to assess the generalization of a pre-training model to unseen distributions, following CLIP.
> As noted by the reviewer, in language-image pre-training, it is almost impossible to avoid all similar annotations as the dataset is annotated with free-form texts. Indeed, we want to benefit from this kind of annotation.
> As GLIP [1] states, we should scale up visual concepts with massive image-text data to ensure a good transferability in language-image pre-training. Even prior to the emergence of CLIP, the use of semantic embeddings and knowledge graph to transfer to zero-shot learning [2] and few-shot learning [3] is also one of many trends.
>
> Secondly, we provide some analysis to give a sense of the verb overlap of HICO with VG. Note that in the table below, we use ``relationship aliases'' [10] to obtain as many HOI verb annotations from VG as possible.
>
> | Dataset | \#images | HOI verb annos | HOI verb annos' ratio | Imbalance ratio |
> | ----- | :-----: | :-----: | :-----: | :-----: |
> | VG | 108K | 2,203 | 1.47\% | 304 |
>
> We can see from the table that in VG, we have only 2,203 HOI verb annotations even when considering relationship aliases, which is about 1.47\% of the number of relationship annotations in HICO-DET. 30 HOI verbs do not have a single annotation, and 45 HOI verbs have 5 or fewer annotations. In RLIP-ParSe (COCO+VG), we observe that mAP for the 30 verbs is 5.56, and mAP for the remaining 87 verbs is 18.12. If we use a uni-modal relation detection pre-training, the result for the 30 verbs degrades to zero.
> In light of this, we conjecture that existing relations can transfer their knowledge to the inference of non-existing relations in HOI detection.
>
>
> >**Q3**: In Eq. 5, is the Euclidean distance the best choice? How about the others like cosine distance?
>
> **A3**: During the design of our model, We tried two distance measuring function Euclidean distance and Cosine distance. The zero-shot (NF) results of Cosine distance using RLIP-ParSeD is shown in the table below.
>
> | Distance Metric | $\eta$ | Rare | Non-Rare | Full |
> | ----- | :-----: | :-----: | :-----: | :-----: |
> | Cosine | 0.3 | 11.21 | 12.53 | 12.23 |
> | Cosine | 0.4 | 11.92 | 12.82 | 12.61 |
> | Cosine | 0.5 | 11.76 | 12.71 | 12.49 |
> | Cosine | 0.6 | 11.30 | 12.22 | 12.01 |
> | Euclidean | 0.3 | 12.30 | 12.81 | 12.69 |
>
> We observe that Euclidean distance is slightly better (the last row of results is selected in the paper). Since both methods have similar computational overhead, in the paper, we choose the Euclidean distance and provide the sensitivity analysis of $\eta$ in the Supplementary Material.

---

> > ### Comment · Reviewer_MVep · 2022-08-03
> > **Post-rebuttal-1**
> >
> > Thanks for the detailed responses from the authors. I will first respond to my concerns one by one:
> >
> > Q1: Addressed. Thanks, please add this clarification if possible in the main text or suppl.
> >
> > Q2: Thanks for the comparison between the labels from the two datasets. CLIP paves a new way for our community, it is indeed hard to compare what the model sees in training and whether this will affect the inference in transfer learning. However, this does not mean we do not need to consider this problem. Recently, many analysis papers are also proposed to see what CLIP or similar big models learn in the training with mixed large-scale datasets. I still believe it is essential to give this comparison and corresponding discussion about the possible data leak in the paper.n The reason is very simple: we should know what these v-l models learn and why they perform well, is the improvement from the increasing training data with possible label population or the fancy general representation? And what we can do in the future to go further, as the collection of more data is harder and harder. Besides, some works also find that CLIP-style works usually have an obvious bias toward text more than visual images/videos. This is very interesting and is also related to my question.
> >
> > Q3: thanks for the response, please add it to the ablations.

---

### Official Review · Reviewer_4wiK · 2022-07-11

**Rating:** 5
**Confidence:** 5
**Soundness:** 2 fair
**Presentation:** 3 good
**Contribution:** 2 fair

**Summary:**

This paper aims to adopt the language-image contrastive pre-training techniques to boost the performance and robustness of the Human-object Interaction (HOI) detection task. For this purpose, this paper first modifies the conventional DETR-based HOI detection framework by decoupling detection and interaction classification and disentangling the subject and object queries. Then, this paper converts the entity labels and relation labels into text and embeds them into a latent space, and constructs contrastive pairs in or out of batches. Extensive experiments have been conducted in HICO-Det and V-COCO datasets from regular, few-shot, and zero-shot settings.

**Questions:**

- My main concerns and questions lie in the 'Missing important baseline' in the weaknesses. The author should provide the results with the mentioned setting for a fair comparison to verify the effectiveness of the proposed RLIP.

- Confused by the OD+VG performances. VG is a large-scale visual relationship detection dataset, but the performance has dropped a lot when pre-training in this dataset. A good choice may be to train DETR in VG first and then train in COCO because the object labels of HICO-Det are the same as COCO. A detailed analysis of VG should be included.

**Limitations:**

The authors provided limitations in supplementary materials.

**Strengths And Weaknesses:**

Strengths:
- The idea and motivation of this paper are generally reasonable.
- The proposed method can be easy to adapt zero-shot, few-shot, and regular HOI detection tasks.
- The writing of this paper is generally clear and the proposed method is simple and easy to follow.

Weaknesses
- The proposed architecture is not new in the HOI detection area, which combines the CDN [1] and PST [2], and the idea is entirely the same as the proposed architecture GEN in the recent paper GEN-VLKT [3]. Though GEN-VLKT is a CVPR 2022 paper, it has been uploaded in Arxiv and released code in Mar. 2022. Thus the author may present a detailed discussion with this paper.

- Limited Application Scenarios. The proposed pre-training techniques still require relation triplets annotations, which are expensive to obtain.

- Missing details about OD+VG. The DETR requires a long time to converge, so the author should report the number of epochs to pre-train the DETR in VG. The significant performance gap between OD and RLIP with VG may come from the insufficient pre-training for OD+VG.

- Missing important baseline. The OD+VG baseline only trains a detector in VG with the detection annotation, but RLIP adopts the relation and detection annotations in the VG simultaneously. Thus a fair baseline is to directly to pre-train the ParSe with the VG visual relationship detection based on the COCO pre-trained model.

- An additional large-scale dataset with a minor improvement. Equipped with COCO+VG pre-training, the performance only improved by 1.05 mAP (31.79->32.84).

[1] Mining the Benefits of Two-stage and One-stage HOI Detection. NIPS 2021.
[2] Visual Relationship Detection Using Part-and-Sum Transformers with Composite Queries. ICCV 2021.
[3] GEN-VLKT: Simplify Association and Enhance Interaction Understanding for HOI Detection. Arxiv Mar. 2022.

---

> ### Author Response · Authors · 2022-08-02
> **Response to Reviewer 4wiK (Part 3/3)**
>
> >**Q4**: Missing important baseline.
>
> **A4**: To verify the effectiveness of the proposed RLIP, we provide further results by performing relation detection pre-tarining on VG. The results are shown in the table below. (PT denotes pre-training.)
>
> | Method | Detector | Data | PT Paradigm | PT \#Epochs | Rare | Non-Rare | Full |
> | ----- | :-----: | :-----: | :-----: | :-----: | :-----: | :-----: | :-----: |
> | ParSeD | DDETR | VG | OD | 50 | 19.59 | 25.03 | 23.78 |
> | ParSeD | DDETR | VG | Relation Detction | 50 | 21.36 | 29.27 | 27.45 |
> | RLIP-ParSeD | DDETR | VG | RLIP | 50 | 24.45 | 30.63 | 29.21 |
> | RLIP-ParSeD | DDETR | COCO+VG | RLIP | 50 |24.67 | 32.50 | 30.70 |
>
> Comparing OD and relation detection pre-training, we observe relatively better performance of relation detection pre-training (23.78->27.45) because of the usage of relation annotations. It can also be interpreted as a more aligned pre-training with the downstream task. However, it's still inferior to RLIP (27.45<29.21). RLIP and relation detection pre-training both incorporate relations as a pre-training signal, while the latter treats every class as a one-hot vector and optimize the model with one-of-N objectives for both the verbs and entities. This paradigm can be sub-optimal due to the semantic similarity of the free-form text labels [8]. Besides, we can not ignore the fact that 30 verbs do not appear in the VG dataset. Thus, we may fail to perform zero-shot (NF) evaluation with uni-modal relation detection pre-training during application, which proves the importance and practicality of RLIP.
>
> Apart from the above experiments, we provide more results with relation detection pre-training upon COCO initialization.
>
> | Method | Detector | Data | PT Paradigm | PT \#Epochs | Rare | Non-Rare | Full |
> | ----- | :-----: | :-----: | :-----: | :-----: | :-----: | :-----: | :-----: |
> | ParSe | DETR | COCO+VG | Relation Detction | 150 | 26.00 | 33.40 | 31.70 |
> | RLIP-ParSe | DETR | COCO+VG | RLIP | 150 | 26.85 | 34.63 | 32.84 |
>
> We can see from the Table that even when providing a good foundation for object detection, adopting RLIP still surpasses relation detection pre-training, which further demonstrates the usefulness of RLIP.
>
> **Reference**:
> [1] GEN-VLKT: Simplify Association and Enhance Interaction Understanding for HOI Detection, submitted to arXiv on Mar. 26th 2022.
> [2] Mining the Benefits of Two-stage and One-stage HOI Detection, NeurIPS 2021.
> [3] Visual Relationship Detection Using Part-and-Sum Transformers with Composite Queries, ICCV 2021.
> [4] Revisiting the sibling head in object detector, CVPR 2020.
> [5] Ppdm: Parallel point detection and matching for real-time human-object interaction detection, CVPR 2020.
> [6] spaCy: Industrial-strength Natural Language Processing in Python, GitHub.
> [7] Grounded language-image pre-training, CVPR 2022.
> [8] Learning transferable visual models from natural language supervision, ICML 2021.
> [9] https://visualgenome.org/api/v0/api_home.html
> [10] Equalization loss for long-tailed object recognition, CVPR 2020.
> [11] Scene graph generation by iterative message passing, CVPR 2017.

---

> > ### Comment · Reviewer_4wiK · 2022-08-03
> > **Responses for rebuttals**
> >
> > Thanks for your response. The experimental results and analysis based on the VG relationship pre-training have verified the effectiveness of the proposed RLIP module and sufficiently solved my main concerns. The authors may replace the original results with VG detection pre-training with such results for a fair comparison in the revised version. However, the concern about the novelty of ParSe remains unsolved for me. Though some implementation details differ from GEN, the core idea is the same. Therefore, I tend to change my rating from 4 BR to 5 BA.

---

> > > ### Author Response · Authors · 2022-08-05
> > > **Response to Reviewer 4wiK**
> > >
> > > Thank you for raising your score and your appreciation for RLIP.
> > > We have made several revisions to the main paper and Supplementary Material as you suggested:
> > >
> > > 1. we add pre-training details (epochs) for object detection and relation detection pre-training;
> > > 2. we add relation detection pre-training results on HICO-DET and V-COCO in Table 1;
> > > 3. we add few-shot transfer results with relation detection pre-training on HICO-DET in Table 3;
> > > 4. we add GEN-VLKT into the related work;
> > > 5. we polish the limitation part in the Supplementary Material to include potential research directions to scale up datasets and boost performance.
> > >
> > > With respect to the remaining concern about ParSe, we want to emphasize the starting point of RLIP, which leads to the design departure from GEN-VLKT. GEN-VLKT aims to simplify association, thus Position Guided Embedding and two groups of queries are designed to index queries and decode for entities; also, it aims to transfer knowledge from CLIP, thus an object-verb coupled classifier is designed to inject textual representations augmented by manual prompts and distill the knowledge from the CLIP visual encoder, **while RLIP targets a language-image pre-training method aligned with HOI detection**. In light of this idea, we want to achieve fine-grained cross-modal alignment (i.e., matching each textual concept with corresponding visual representations rather than matching one image with one sentence), which can be beneficial to a detection task [1,2]. Thus, we design ParSe to facilitate the instantiation of RLIP. The entity-level cross-modal alignment is more flexible and apt to expand to large-scale triplets. We hope these two research directions can both motivate further research.
> > >
> > > Thanks again for your sincere suggestions and your timely follow-ups.
> > >
> > > **Reference**:
> > > [1] Grounded language-image pre-training, CVPR 2022.
> > > [2] X-DETR: A Versatile Architecture for Instance-wise Vision-Language Tasks, ECCV 2022.

---

> ### Author Response · Authors · 2022-08-02
> **Response to Reviewer 4wiK (Part 2/3)**
>
> >**Q2**: Limited Application Scenarios.
>
> **A2**: This is the first work to directly incorporate relations as a language-image pre-training signal, which underpins further relational pre-training research and motivates more application scenarios with scarce data and free-form text inputs. Although existing relation annotations are limited (as we note in limitations), we do not anticipate that this will remain the case. Indeed, we hope that our work will inspire future work to focus on this problem and dataset contributions will follow.
> Besides, we provide ways to scale up datasets as future works.
> For example, we could reuse a grounding dataset with entities annotated. Then, a language processing tool like spaCy [6] can be adopted as a tool to obtain their relations from captions. Even if we do not have subjects and objects but only image-caption pairs, we can combine the use of spaCy and methods like GLIP [7] to create abundant triplet annotations. Based on the analysis, we think our method is still promising and inspiring, paving a path for further research.
>
>
> >**Q3**: Missing details about OD+VG, and being confused about its performance since the performance has dropped a lot when performing OD pre-training on VG.
>
> **A3**: In the paper, to accelerate the convergence of a detection model, we use Deformable DETR (DDETR) as a base detector to perform this series of experiments and use the common training settings (50 epochs).
>
> The ability of relation detection is comprised of two parts: object detection ability and relation inference ability.
> The widely-adopted COCO pre-training contributes to a good object detection foundation (abundant object annotations with identical classes to HICO-DET and VCOCO), thus previous methods can perform well.
> However, the relation inference ability is under-explored.
>
> VG dataset is adopted for its potential relation inference foundation, though it's annotated with very noisy free-form texts.
> By only performing OD pre-training on VG, we try to uncover the object detection foundation of VG for HOI detection.
> According to the results in the table below, we can see that VG has an inferior object detection foundation for HOI downstream tasks (note: PT stands for pre-training).
>
> | Method | Detector | Data | PT Paradigm | PT \#Epochs | Rare | Non-Rare | Full |
> | ----- | :-----: | :-----: | :-----: | :-----: | :-----: | :-----: | :-----: |
> | ParSeD | DDETR | VG | OD | 50 | 19.59 | 25.03 | 23.78 |
> | ParSeD | DDETR | COCO | OD | 50 | 22.23 | 31.17 | 29.12 |
> | RLIP-ParSeD | DDETR | VG | RLIP | 50 | 24.45 | 30.63 | 29.21 |
> | RLIP-ParSeD | DDETR | COCO+VG | RLIP | 50 |24.67 | 32.50 | 30.70 |
>
>
> We give more statistical data as an intuitive observation. Note that in the table below, we use ``object aliases'' [9] to obtain as many HOI object annotations from VG as possible.
>
> | Dataset | \#images | HOI object annos | Imbalance ratio |
> | ----- | :-----: | :-----: | :-----: |
> | COCO | 118K | 860K | 1308 |
> | VG | 108K | 518K | 69318 |
>
> We can conclude from the table that COCO has more images, more HOI object annotations, and a much smaller annotation imbalance ratio [10].
> Besides, the object annotations in VG are relatively of poor quality [11].
> Furthermore, due to the semantic ambiguity of free-form texts, one-of-N objectives are sub-optimal to optimize the OD detection model on VG.
> Considering all the factors mentioned above, we can conjecture that the low HOI performance of OD pre-training on VG is reasonable.
>
> The COCO+VG model is a simple tryout to transfer part of the object detection ability in COCO to RLIP on VG, serving as a remedy for VG's relatively low object detection foundation.
> Further research can focus on more elaborate transferring methods, which can be formulated as a semi-supervised task.

---

> ### Author Response · Authors · 2022-08-02
> **Response to Reviewer 4wiK (Part 1/3)**
>
> We thank the reviewer for their valued feedback and are encouraged that they find our idea and motivation reasonable, our method easily adaptable to various settings and our writing generally clear. To address their concerns, we present more analysis to compare to previous work (Q1), propose methods to potentially address limited application scenarios (Q2), provide more details and observations to clarify the rationality of performance (Q3) and provide thorough baselines to demonstrate the usefulness of RLIP (Q4).
>
>
> >**Q1**: Comparisons with GEN-VLKT [1], CDN [2] and PST [3].
>
> **A1**: We thank the reviewer for highlighting this reference. As noted by the reviewer, GEN-VLKT represents concurrent work (GEN-VLKT is submitted to arXiv on March 26th 2022 while the NeurIPS abstract deadline was 16th May, 2022.). We will include this in our related work and clarify our differences here.
>
> While GEN-VLKT mimics the image representations from CLIP, RLIP starts from **a different perspective to directly transfer relation inference ability** from a dataset annotated with free-form relation texts to downstream tasks. Compared to CLIP pre-trained on 400 million image-text pairs, our model utilizes a much smaller dataset (108K) to perform language-image pre-training at the level of entities and relations to align with the downstream task. It can potentially be combined with GLIP [7] to scale up. The proposed aligned pre-training enables the model to have the ability of zero-shot HOI detection without any fine-tuning, exhibit good performance when data is scarce and be robust towards relation label noise. When considering only the architectural designs employed for triplet detection, ParSe and GEN share similar architectures but also have several differences: **i)** when transferring knowledge, ParSe targets an HOI classifier that is relation-object disentangled, aiming to align with the decoupled representations and the contrastive loss, which enables an extension to identify as many combinations as possible. By contrast, GEN couples the relations and objects into an interaction classifier in order to utilize the off-the-shelf CLIP text encoder; **ii)** To infer relations, ParSe iteratively decodes relation features for the last layer's output from Parallel Entity Inference for all the Sequential Relation Inference layers, while GEN decodes relations for every subject and object pair from Instance Decoder by one-layer decoding; **iii)** ParSe does not have Position-Guided Embeddings. We believe that the index-matching adopted by ParSe can optimize the queries to decode for related subjects and objects.
>
> With respect to comparisons with CDN, RLIP is motivated by language-image pre-training. The triplet detection structure ParSe builds on CDN as stated in the paper and further improves upon it with more decoupled representations for better contrastive learning, because
> **i)** from a perspective of language-image pre-training, we can align the representations of individual entities to their corresponding texts;
> **ii)** from a perspective of uni-modal detection, entity detection is position-sensitive [4], and thus, avoiding multi-task learning can improve the performance [2].
>
> With respect to PST, its overall design follows PPDM [5] with a DETR instantiation, which performs parallel decoding for Sum and Part queries, while ParSe is a sequential structure. Due to PST's parallel structure, PST has to design a factorized self-attention layer to further enhance part-level learning and extra modules to enhance part-sum interaction. In comparison, ParSe is simpler (with Parallel Entity Inference and Sequential Relation Inference). Also, the generation of relation queries differs significantly because in ParSe, relation queries are conditionally generated by subjects and objects, which injects stronger priors into the relation inference stage. ParSe shows much stronger performance (+7.86 mAP on Full set) over it.
>
> All in all, we think RLIP represents a substantial departure from existing work such as CDN and PST because of the new pre-training signal and paradigm we propose.
> We believe this which can benefit the research community, not limited to HOI but also VQA or visual reasoning where relations can potentially help.

---

### Official Review · Reviewer_7c46 · 2022-07-13

**Rating:** 6
**Confidence:** 3
**Soundness:** 4 excellent
**Presentation:** 2 fair
**Contribution:** 3 good

**Summary:**

For Human-Object Interaction (HOI) detection, the authors propose Relational Language-Image Pre-training (RLIP), a strategy for contrastive pre-training that leverages both entity and relation descriptions.
To make effective use of such pre-training, they make three technical contributions:
(1) a new Parallel entity detection and Sequential relation inference (ParSe) architecture that enables the use of both entity and relation descriptions during pre-training;
(2) a synthetic data generation framework, Label Sequence Extension, that expands the scale of language data available within each minibatch;
(3) ambiguity-suppression mechanisms, Relation Quality Labels, and Relation Pseudo-Labels, to mitigate the influence of ambiguous/noisy samples in the pre-training data.
Through extensive experiments, they demonstrate the benefits of these contributions, collectively termed RLIP-ParSe, for improved zero-shot, few-shot, and fine-tuning HOI detection performance as well as increased robustness to learning from noisy annotations.


**Questions:**

In Table 1, even without the RLIP paradigm, how do ParSe and ParseD show favorable performance compared to the previous method?

**Limitations:**

The authors addressed the limitations of the proposed method and the potential negative social impact of their work.

**Strengths And Weaknesses:**

Strength

- The authors show the effectiveness of the proposed method on various setups such as few-shot and zero-shot setups.

- The proposed ParSe model consistently shows favorable performance compared to the existing methods.

Weakness

- The original contribution of the ParSe model is unclear. In Sec.3, the authors should further emphasize the technical significance of the proposed ParSe model compared to the existing architectures. From the current explanation, it is hard to see the novelty of the proposed model.

- In Table 1, in order to more comprehensively validate the effectiveness of the proposed pre-training method with external data, it would be better to compare with the existing methods when using VG data when using VG data as well.


========== ------- Comments after the rebuttal ------========

I have read the other reviewer's comments and the author's rebuttal, which addresses most of my concerns. Therefore, I would like to raise my score.

---

> ### Author Response · Authors · 2022-08-02
> **Response to Reviwer 7c46**
>
> We thank reviewer 7c46 for their valued feedback.
> We are encouraged that they find the zero-shot, few-shot and fine-tuning performance significant compared to previous methods.
> To address their concerns, we present more thorough experiments and analysis to show the contribution of ParSe (Q1) and demonstrate the superiority of RLIP even if comparing with previous methods adopting the VG dataset (Q2).
>
> >**Q1**. Technical significance of the proposed ParSe model compared to the existing architectures. How do ParSe and ParseD show favorable performance compared to the previous method without RLIP?
>
> **A1**: The motivation for the design of ParSe is to achieve better language-image contrastive learning.
> For a fine-grained task like HOI detection, we need to align visual representations and textual representations if performing language-image contrastive learning.
> If we use previous methods like QPIC [1] or CDN [2], we need to align visual representations of (or some subset of) the subject, object and relation triplets with their corresponding textual representations, which will result in inferior performance.
> As the representations for the detection task are position-sensitive [3], the model can achieve better performance by avoiding decoded queries to perform multi-task learning.
> Since we use Cross-Entropy loss and Focal loss as contrastive losses, which are identical to the ones we use in fine-tuning the uni-modal ParSe, we can observe the superiority of this design directly by observing the uni-modal fine-tuning results (these also appear in the Supplementary Material).
>
> | ParSe Architecture | Coupling | Rare | Non-Rare | Full |
> | ----- | :----- | :-----: | :-----: | :-----: |
> | - | coupled subject, objects and relations [1] | 23.18 | 31.45 | 29.55 |
> | w/ Se | coupled subject and objects [2] | 25.58 | 32.50 | 30.91 |
> | w/ ParSe | fully decoupled | 26.36 | 33.41 | 31.79 |
>
> We can see from the above table that by ablating the decoupled design itself, we could gradually improve the performance.
> From the perspective of visualizations in Figure 3 in the main paper, we can observe that ParSe attends to distinct regions where appropriate to better represent the target triplets.
>
> >**Q2**. In order to more comprehensively validate the effectiveness of the proposed pre-training method with external data, it would be better to compare with the existing methods when using VG data as well.
>
> **A2**: We think it to be a useful suggestion. To have a fairer comparison with previous methods, we adopt CDN [2] as a base method and then add the VG dataset to its pre-training stage. Since RLIP also adopts the relation annotations in VG, we also try to include these annotations in the uni-modal pre-training. Thus, we resort to relation detection on VG. To be more specific, we perform uni-modal relation detection pre-training by using linear classifiers for verbs and entities rather than matching with texts. The results are shown in the table below.
>
> | Method | Detector | Data | PT Paradigm | PT \#Epochs | Rare | Non-Rare | Full |
> | ----- | :-----: | :-----: | :-----: | :-----: | :-----: | :-----: | :-----: |
> | CDN | DETR | COCO+VG | Relation Detction | 150 | 25.65 | 32.75 | 31.12 |
> | ParSe | DETR | COCO+VG | Relation Detction | 150 | 26.00 | 33.40 | 31.70 |
> | RLIP-ParSe | DETR | COCO+VG | RLIP | 150 | 26.85 | 34.63 | 32.84 |
>
> We can see from the table that by using uni-modal relation detection pre-training, CDN still trails RLIP-ParSe with the same number of epochs of pre-training and fine-tuning, which shows the effectiveness of RLIP. Even if comparing it with ParSe using relation detection pre-training, we can still observe an improvement of ParSe over CDN, demonstrating the usefulness of decoupling triplet representations.
>
> **Reference**:
> [1] Qpic: Query-based pairwise human-object interaction detection with image-wide contextual information, CVPR 2021.
> [2] Mining the benefits of two-stage and one-stage HOI detection, NeurIPS 2021.
> [3] Revisiting the sibling head in object detector, CVPR 2020.

---

> ### Author Response · Authors · 2022-08-06
> **A Gentle Reminder of Feedbacks**
>
> Dear Reviewer 7c46,
>
> Thanks for your careful comments and your appreciation for our work. We have revised our paper and added the experiments and analysis of incorporating VG and COCO datasets for previous methods into the Supplementary Material. Currently, all of your concerns can be resolved in the revised version of the paper and Supplementary Material. We want to leave a gentle reminder that the discussion period is closing. We would appreciate your feedback to make sure that our responses and revisions have resolved your concerns, or whether there is a leftover concern that we can address to ensure a quality work.
>
> Yours sincerely,
> Authors of Paper226

---

> ### Comment · Reviewer_7c46 · 2022-08-09
> **Response**
>
> Thank you, authors, for your elaborate rebuttal. I have read the other reviewer's comments and the author's rebuttal, which addresses most of my concerns. Therefore, I would like to raise my score to 6.

---

### Official Review · Reviewer_VZWS · 2022-07-14

**Rating:** 6
**Confidence:** 4
**Soundness:** 3 good
**Presentation:** 3 good
**Contribution:** 3 good

**Summary:**

-- The paper proposes Relational Language-Image Pre-training(RLIP) a pre-training paradigm for Human-Object Interaction (HOI) Detection. This paper posits that this pre-training methodology aims to overcome the gaps left by Object Detection pre-training methodologies that a not fully tailored towards HoI Detection.

-- The paper presents multiple modules to achieve this pre-training objective. Parallel entity detection and Sequential relation inference (ParSe) proposes a DETR-like training architecture that employs query groups for Subject and Object representation and further query groups conditional on these for modeling Relational representations. To enable contrastive learning during training, the paper proposes  Label Sequence Extension (LSE) that performs out-of-batch-sampling to improve quality of negative samples. Finally, to address label noise and ambiguity in relationship modeling, the paper proposes employing Relational Quality Labels (RQL) and Relational Pseudo-Labels (RPL) modules during training.

-- The paper reports the benefits of the pre-training methodology for HOI detection methods for fine-tuned, zero-shot and few-shot settings.

**Questions:**

-- Did the authors try any dataset other than VG with similar entity description to showcase the robustness of the model?

-- The authors could try to include more qualitative examples in the supplementary material to showcase success and failure cases.

-- There are couple of small changes that could be made in writing

    - #L57 'negatives samples'  -> 'negative samples'
    - #L144 'We next similarly' -> Next, we similarly

**Limitations:**

-- There are no suggestions on the societal impact front --

**Strengths And Weaknesses:**

+ The paper leverages the entity descriptions of Visual Genome (VG) datasets efficiently to improve the performance on HOI detection, one of the challenging but relevant problems for scene understanding.  The proposed ParSE method leverages the relational entities with context along with subject and object entities to improve <subject-object-relation> prediction.

+ The paper proposed modules to refine the label noise and semantic ambiguity in the VG dataset using the RQL and RPL modules which is a relevant problem for VG dataset.


+ The paper presents strong results for the zero-shot HOI detection over other SoTA approaches demonstrating the effectiveness of pre-training method for zero-shot detection. The paper also shows the contribution of each of the modules through ablation studies showcasing the contributions of each of the modules.

- The proposed method does not show a strong improvement over the CDN method for fully-supervised HOI detection method for both VCOC and HICO-DET datasets. Infact, the performance of the method goes down when including COCO dataset for RLIP compared to just the OD pre-training method using ParSE indicating overfitting of the pre-training method when including COCO.

- The improvement for zero-shot setting could be down to the strong prior in the distribution provided by the VG dataset since the zero-shot formulation is based on OOD rather Unseen setting. Including another dataset with triplet entities of Subject, Object, Relation might showcase the robustness of the method to different data distributions.

- The method ranks low in novelty since it reuses many of the existing HOI detection methodologies and adapts the same for pre-training using VG dataset.

---

> ### Author Response · Authors · 2022-08-02
> **Response to Reviwer VZWS (Part 2/2)**
>
> >**Q3**. Did the authors try any dataset other than VG with similar entity description to showcase the robustness of the model?
>
> **A3**: While this is certainly a good suggestion, there are limited datasets available with relation annotation of comparable size.
> We therefore take a different approach to showcase RLIP's robustness across different upstream dataset distributions.
>
> We do this by significantly altering the distribution of VG annotations and assessing the influence on RLIP's performance.
> To this end, first note that since VG is human-annotated with free-form text, it is extremely long-tailed.
> We alter its distribution by dropping tail object classes and verb classes to create a dataset with limited semantic diversity.
> Concretely, we drop object classes whose instance counts are fewer than 1,000 and relation classes whose instance counts are fewer than 500.
> We pre-train RLIP on the resulting dataset and then perform zero-shot (NF) evaluation on HICO-DET.
> The results are shown in the Table below. (Obj, rel and annos denote object, relation and annotations respectively.)
>
> | Method | Data | Obj classes | Obj annos | Rel classes | Rel annos | Rare | Non-Rare | Full |
> | ----- | :-----: | :-----: | :-----: | :-----: | :-----: | :-----: | :-----: | :-----: |
> | ParSeD | VG | 100,298 | 3.80m | 36,515 | 1.99m | 12.30 | 12.81 | 12.69 |
> | ParSeD | VG- |    497 | 1.73m |    151 | 1.27m |  9.45 | 12.13 | 11.51 |
>
> We observe from this table that despite a very significant change to the training distribution, performance on the Full set drops only moderately.
> We do, however, witness a relatively larger decline on the Rare set due to the lack of semantic diversity in the modified data.
> This finding accords with the observations of GLIP [3].
> To make full use of language-image pre-training, semantic diversity is important which can ensure a good domain transfer as is indicated by CLIP and GLIP [3,4].
>
>
> **Reference**:
> [1] Qpic: Query-based pairwise human-object interaction detection with image-wide contextual information, CVPR 2021.
> [2] Mining the benefits of two-stage and one-stage HOI detection, NeurIPS 2021.
> [3] Grounded language-image pre-training, CVPR 2022.
> [4] Learning transferable visual models from natural language supervision, ICML 2021.

---

> ### Author Response · Authors · 2022-08-02
> **Response to Reviwer VZWS (Part 1/2)**
>
> We thank reviewer VZWS for their valued comments.
> We are encouraged that they find our work novel.
> To address their concerns, we present more thorough experiments and analysis to compare with previous methods (Q1), showcase the robustness of the model (Q3), and clarify the performance improvement (Q2).
> We will also include more failure cases in the Supplementary Material as suggested.
>
>
> >**Q1**: The proposed method does not show a strong improvement over the CDN method for fully-supervised HOI detection method for both VCOCO and HICO-DET datasets.
>
> **A1**: The design of ParSe aims to facilitate Relational Language-Image Pre-training by decoupling the representations of <subject, relation, object> triplets.
> This decoupled design is also beneficial to uni-modal HOI detection.
> To see this, we can compare with CDN [1] using the same number of decoder layers, where we observe that ParSe outperforms CDN across different backbones.
>
> | Method | Backbone | PT Data | \#Tuning epochs | Rare | Non-Rare | Full |
> | ----- | :-----: | :-----: | :-----: | :-----: | :-----: | :-----: |
> | CDN | ResNet-50 | COCO | 90+10 | 27.39 | 32.64 | 31.44 |
> | ParSe | ResNet-50 | COCO | 90 | 26.36 | 33.41 | 31.79 |
> | CDN | ResNet-101 | COCO | 90+10 | 27.19 | 33.53 | 32.07 |
> | ParSe | ResNet-101 | COCO | 90 | 28.59 | 34.01 | 32.76 |
>
> To better understand how this decoupled design helps uni-modal HOI detection, we also ablate the design itself (note that CDN uses dynamic re-weighting as a two-stage tuning method---this is a good design but we focus on the model design here), the result of which is shown below (also in the Supplementary Material).
>
> | ParSe Architecture | Coupling | Rare | Non-Rare | Full |
> | ----- | :----- | :-----: | :-----: | :-----: |
> | - | coupled subject, objects and relations [2] | 23.18 | 31.45 | 29.55 |
> | w/ Se | coupled subject and objects [1] | 25.58 | 32.50 | 30.91 |
> | w/ ParSe | fully decoupled | 26.36 | 33.41 | 31.79 |
>
> We can see that by decoupling the representations, we boost performance.
>
> With respect to the performance on V-COCO, if we use COCO pre-training, we can have a clear gain (61.7 $\rightarrow$ 62.5 in Scenario1 and 63.8 $\rightarrow$ 64.8 in Scenario2).
> However, when using RLIP on VG, the gain is small.
> We attribute this to the reduced domain alignment of COCO pre-training, because V-COCO is a dataset based on COCO images, thus using COCO pre-training is more favorable than RLIP on VG (this observation is noted in Line 284).
>
> >**Q2**. In fact, the performance of the method goes down when including COCO dataset for RLIP compared to just the OD pre-training method using ParSe indicating overfitting of the pre-training method when including COCO.
>
> **A2**: To clarify any potential misunderstanding, we report results below.
> When using COCO with RLIP, performance always surpasses using OD pre-training on COCO.
> The comparison is shown in the Table below.
>
> | Data | PTP | Method | Detector | Rare | Non-Rare | Full |
> | ----- | :-----: | :-----: | :-----: | :-----: | :-----: | :-----: |
> | COCO | OD | ParSeD | DDETR | 22.23 | 31.17 | 29.12 |
> | COCO+VG | RLIP | RLIP-ParSeD | DDETR | 24.67 | 32.50 | 30.70 |
> | COCO | OD | ParSeD | DDETR | 26.36 | 33.41 | 31.79 |
> | COCO+VG | RLIP | RLIP-ParSeD | DDETR | 26.85 | 34.63 | 32.84 |
>
> We can see that when we use the same base detector, RLIP can benefit from COCO initialization (potentially because COCO initialization provides a good object detection foundation).
> This gap widens when data becomes more scarce (i.e. in the few-shot setting).

---

> ### Author Response · Authors · 2022-08-07
> **A Gentle Reminder of Feedbacks**
>
> Dear Reviewer VZWS,
>
> Thanks for your careful comments and your appreciation for our work. We have revised our paper and added the experiments and analysis concerning
> 1. more thorough comparisons with CDN in Table 2 of the Supplementary Material, with corresponding analysis starting from Line 99 to Line 103;
> 2. the influence of upstream data distributions in Table 14 of the Supplementary Material, with corresponding analysis starting from Line 224 to Line 236;
> 3. more successful and failure case visualization, and potential future works in Figure 5 of the Supplementary Material, with corresponding analysis starting from Line 237 to Line 253;
>
> Currently, all of your concerns can be resolved in the revised version of the paper and Supplementary Material. We want to leave a gentle reminder that the discussion period is closing. We would appreciate your feedback to make sure that our responses and revisions have resolved your concerns, or whether there is a leftover concern that we can address to ensure a quality work.
>
> Yours sincerely,
> Authors of Paper226

---

### Author Response · Authors · 2022-08-09
**Summary of the Rebuttal and Discussions**

Dear Chairs and Reviewers,

As the discussion period comes to an end, we want to present a brief summary of our rebuttal and discussions with the reviewers for further reference.

First of all, we thank all reviewers for their efforts and valuable comments. We are encouraged that the reviewers found RLIP shows strong zero-shot and few-shot performance (Reviewer VZWS, Reviewer 7c46, Reviewer 4wiK), has a non-trivial idea/motivation (Reviewer 4wiK, Reviewer MVep), has thorough experiments&analysis (Reviewer MVep) and is well written (Reviewer 4wiK). We are also glad to receive positive feedback from reviewers that our responses and paper revision have addressed their concerns.
_____________________
Secondly, building on the rebuttal and discussions, we summarize this paper RLIP as follows:
- **Observation:** Prior works have demonstrated the benefits of effective architecture design and integration of relevant cues for HOI detection.
Among all these methods, object detection had been a de-facto  pre-training paradigm for HOI detection, while they under-explored the pre-training of relation inference ability.
- **Motivation:** We could leverage the dataset annotated with free-form texts, use relations as a pre-training signal, and transfer the relation inference ability to downstream tasks in various settings.
- **Methodology:** We propose RLIP for transferring relation inference ability. To be more specific, we propose ParSe to facilitate fine-grained entity- and relation-level contrastive learning, a synthetic language data generation framework to improve contrastive learning, and mechanisms to account for relational semantic ambiguity and noise.
- **Experiments:** Extensive experiments and analysis are performed to prove RLIP's superiority over object detection, relation detection and modulated detection [1] pre-training, and its consistent boost for zero-shot, few-shot and fine-tuning HOI detection performance as well as increased robustness to learning from noisy annotations.
- **Future Works:** As CLIP [2] became the milestone for image classification/retrieval and GLIP [3] for object detection, we expect future works will follow RLIP to improve upon it. We expect RLIP can scale up in a semi-supervised manner combined with GLIP. Moreover, we expect the incorporation of relational pre-training could also benefit visual question answering [4] and visual reasoning [5,6], where relation inference counts.

_____________________
Thirdly, we make a brief summary of the key revisions that we made to the main paper and Supplementary Material (SuppM), and detail how they resolve reviewers' concerns:

**[Additional experiments and analysis]**
- **Results with uni-modal relation detection pre-training** in fully-finetuned and few-shot settings in Table 1 and Table 3 of the main paper and also in Table 3 of the SuppM, demonstrating RLIP's superiority over it using the same VG dataset and annotations;
- **Comparison with previous zero-shot methods using COCO for pre-training** in Table 2 of the main paper, proving the usefulness of ParSe and RLIP in zero-shot setting;
- **Comparison with previous methods using different backbones and pre-training datasets** in Table 2 and Table 3 of the SuppM, proving ParSe's superiority across various backbones and pre-training datasets;
- **Detailed zero-shot performance, and qualitative and quantitative analysis for reasons why RLIP can perform zero-shot HICO detection** from Line 164 to Line 223 of the SuppM;
- **RLIP's robustness towards upstream data distributions (semantic diversity)** in Table 14 of the SuppM;
- **Similarity analysis between in-batch labels and out-of-batch labels** in Table 10 of the SuppM, proving the usefulness of contrastive learning for relations and entities;
- **Design choice of distance function in RPL** in Table 8 of the SuppM;
- **Successful and failed case analysis, and corresponding future work** from Line 237 to Line 253 of the SuppM;
- **Potential future works to scale up dataset size and boost performance** in Sec A.2 of the SuppM.

**[More descriptive revisions]**
We revised the main paper to have a more precise contribution claim of ParSe, add more pre-training details for object detection and relation detection, add related works and correct several typos.

_____________________
Thanks again for their efforts to review and give further feedback on RLIP, enabling us to improve our submission.

Yours sincerely,
Authors of Paper226

**Reference**:
[1] MDETR-modulated detection for end-to-end multi-modal understanding, ICCV 2021.
[2] Learning transferable visual models from natural language supervision, ICML 2021.
[3] Grounded language-image pre-training, CVPR 2022.
[4] Vqa: Visual question answering, CVPR 2015.
[5] CLEVR: A Diagnostic Dataset for Compositional Language and Elementary Visual Reasoning, CVPR 2017.
[6] Gqa: A new dataset for real-world visual reasoning and compositional question answering, CVPR 2019.

---

### Meta-Review · Area_Chair_cc8E · 2022-08-26

**Recommendation:** Accept
**Confidence:** Certain

**Metareview:**

This paper proposes a free-form relational language-image pretraining for HOI detection which demonstrates advantageous performances on zero-shot and few-shot settings. All reviewers give consistent positive scores after the discussion phase. The authors have added more experiments on COCO, different backbones, and pretraining datasets. And additional quantitative and qualitative analyses to claim motivations are also presented. Given the good insights and performance of the proposed RLIP, the meta-reviewers thus recommend to accept this paper.

**Award:**

No

---

### Decision · Program_Chairs · 2022-09-14

Accept